# Temperature effect on seawater $f\mathrm{CO_2}$ revisited: theoretical basis, uncertainty analysis, and implications for parameterising carbonic acid equilibrium constants

Matthew P. Humphreys

Department of Ocean Sciences, NIOZ Royal Netherlands Institute for Sea Research, PO Box 59, 1790 AB Den Burg (Texel), the Netherlands

*Correspondence to*: Matthew P. Humphreys (matthew.humphreys@nioz.nl)

**Abstract.** The sensitivity of the fugacity of carbon dioxide in seawater ($f\mathrm{CO_2}$) to temperature (denoted $\upsilon$, reported in % °C$^{-1}$) is critical for the accurate $f\mathrm{CO_2}$ measurements needed to build global carbon budgets and for understanding the drivers of air-
10 sea $\mathrm{CO_2}$ flux variability across the ocean. Yet understanding and computing $\upsilon$ have been restricted to either using empirical functions fitted to experimental data or determining it as an emergent property of a fully resolved marine carbonate system, and these two approaches are not consistent with each other. The lack of a theoretical basis and an uncertainty estimate for $\upsilon$ has hindered resolving this discrepancy. Here, we develop a new approach to calculating the temperature sensitivity of $f\mathrm{CO_2}$ based on the equations governing the marine carbonate system and the van 't Hoff equation. This shows that, to first order,
$\ln(f\mathrm{CO_2})$ should be proportional to $1/t_\mathrm{K}$ (where $t_\mathrm{K}$ is temperature in K), rather than to temperature as has previously been assumed. This new approach is to first order consistent with calculations from a fully resolved marine carbonate system, which we have incorporated into the PyCO2SYS software. Agreement with experimental data is less convincing but remains inconclusive due to the scarcity of direct measurements of $\upsilon$, particularly above 25 °C. However, the new approach is consistent with field data, performing better than any other approach for adjusting $f\mathrm{CO_2}$ by up to 10 °C if spatiotemporal variability in
its single fitted coefficient is accounted for. The uncertainty in $\upsilon$ arising from only measurement uncertainty in the main experimental dataset where $\upsilon$ has been directly measured is on the order of 0.04% °C$^{-1}$, which corresponds to a 0.04% uncertainty in $f\mathrm{CO_2}$ adjusted by +1 °C. However, spatiotemporal variability in $\upsilon$ is several times greater than this, so the true uncertainty due to the temperature adjustment in $f\mathrm{CO_2}$ adjusted by +1 °C using the most widely used constant $\upsilon$ value is around 0.24%. This can be reduced to around 0.06% by using the new approach proposed here, and this could be further reduced by
more measurements. The spatiotemporal variability in $\upsilon$ arises mostly from the equilibrium constants for $\mathrm{CO_2}$ solubility and carbonic acid dissociation ($K_1^*$ and $K_2^*$) and its magnitude varies significantly depending on which parameterisation is used for $K_1^*$ and $K_2^*$. Seawater $f\mathrm{CO_2}$ can be measured accurately enough that additional experiments should be able to detect spatiotemporal variability in $\upsilon$ and distinguish between different parameterisations for $K_1^*$ and $K_2^*$. Because the most widely used constant $\upsilon$ was coincidentally measured from seawater with roughly global average $\upsilon$, our results are unlikely to
significantly affect global air-sea $\mathrm{CO_2}$ flux budgets, but may have more important implications for regional budgets and studies that adjust by larger temperature differences.

## 1 Introduction

The surface ocean continuously exchanges carbon dioxide ($\mathrm{CO_2}$) with the atmosphere. The net direction of exchange varies through space and time, with some ocean areas acting as $\mathrm{CO_2}$ sources and others as sinks (Takahashi et al., 2009). Whether an
35 area acts as a $\mathrm{CO_2}$ source or sink depends on the difference between the fugacity of $\mathrm{CO_2}$ ($f\mathrm{CO_2}$) in the sea and in the air (denoted $\Delta f\mathrm{CO_2} = f\mathrm{CO_2}(\mathrm{sea}) - f\mathrm{CO_2}(\mathrm{air})$), with net $\mathrm{CO_2}$ transfer from high to low $f\mathrm{CO_2}$. The fugacity of $\mathrm{CO_2}$ is similar to its partial pressure ($p\mathrm{CO_2}$) but with a small correction for the non-ideal behaviour of $\mathrm{CO_2}$ in air; $f\mathrm{CO_2}$ is typically about 99.7% of $p\mathrm{CO_2}$. The gas exchange rate is modulated by a set of processes that are commonly parameterised as a function of wind

speed (Wanninkhof, 2014). Most of the surface ocean is either a $CO_2$ source or sink, rather than being in equilibrium with the atmosphere, because many of the processes that modify seawater $fCO_2$ (e.g., seasonal cycles of temperature and primary productivity) operate on faster timescales than air-sea $CO_2$ equilibration, which takes from months to a year for the surface mixed layer (Jones et al., 2014).

On top of the natural background variability, the global ocean is a net $CO_2$ sink due to rising atmospheric $fCO_2$. About a quarter of the anthropogenic $CO_2$ emitted each year, some $2.8 \pm 0.4$ Gt-C in 2022, is taken up by the ocean, mitigating the greenhouse effect of $CO_2$ on Earth's climate (Friedlingstein et al., 2023). Together, uptake of $CO_2$ and warming drive changes in the marine carbonate system, altering biologically important variables like seawater pH and the saturation states with respect to carbonate minerals (Humphreys, 2017). Quantifying marine $CO_2$ uptake requires highly accurate $fCO_2$ measurements: it is driven by a global mean $\Delta fCO_2$ of $-4.1 \pm 1.6$ µatm (Fay et al., 2024), which is around 2% of mean $fCO_2$. Consequently, small systematic biases in seawater $fCO_2$ propagate into large errors in $\Delta fCO_2$ and thus the total ocean $CO_2$ sink. The required measurement accuracy has been quantified by the Global Ocean Acidification Observing Network (GOA-ON), who suggested a minimum accuracy of 0.5% (i.e., around 2 µatm) for "climate quality" seawater $fCO_2$ measurements that can be used to detect decadal trends or 2.5% (~10 µatm) for "weather quality" $fCO_2$ data for investigating changes of greater magnitude, such as seasonal cycles (Newton et al., 2015).

The observational constraint on marine $CO_2$ uptake is based on measurements of surface ocean $fCO_2$ from ships, compiled into data products such as the Surface Ocean $CO_2$ Atlas (SOCAT; Bakker et al., 2016). Surface seawater is equilibrated with an enclosed headspace, which flows through to a measuring device, and the raw measurements are passed through a series of processing steps to obtain seawater $fCO_2$. One important processing step is the temperature adjustment. While it is being pumped from the surface ocean to the equilibration chamber, seawater may warm up or cool down. As $fCO_2$ is highly sensitive to temperature (Weiss, 1974), each measurement must be adjusted back to its in situ temperature in order to obtain environmentally relevant values of $fCO_2$ (Pierrot et al., 2009; Pfeil et al., 2013; Goddijn-Murphy et al., 2015). For the data in SOCAT with the most stringent quality control flags (A and B), a maximum change in temperature ($\Delta t$) of 1 °C between intake and measurement is permitted (although in practice, adjustments are usually around 0.2-0.3 °C; Wanninkhof et al., 2022), in order to minimise uncertainty in $fCO_2$ due to the temperature adjustment. Adjustments with greater $\Delta t$ are often used in other applications. Some groups measure $fCO_2$ in discrete samples at a constant temperature, which must then be adjusted by up to 20 °C to return to in situ conditions (Wanninkhof et al., 2022). Several studies have normalised seawater $fCO_2$ observations to a constant temperature in order to investigate the non-thermal (i.e., chemical) drivers of $fCO_2$ change on various timescales (e.g., Takahashi et al., 2002, 2003, 2006; Feely et al., 2006). Others have computed the temperature sensitivity in order to isolate the thermal component of observed $fCO_2$ change and unravel the drivers of $fCO_2$ variability (e.g., Metzl et al., 2010; Fassbender et al., 2022; Rodgers et al., 2023). Data from SOCAT have been adjusted from the intake temperature, typically measured a few metres below the sea surface, to the temperature directly at the sea surface in order to assess how the "cool skin" effect affects computed air-sea $CO_2$ fluxes (Dong et al., 2022; e.g., Watson et al., 2020). Finally, climatological $fCO_2$ datasets have been adjusted to match different sea surface temperature fields for specific analyses (e.g., Kettle et al., 2009; Land et al., 2013) and this functionality is part of published data processing toolkits (e.g., Shutler et al., 2016; Holding et al., 2019).

The most widely used approach to adjust $fCO_2$ data to different temperatures ($t$), used in all the examples above, is based on data from Takahashi et al. (1993), who measured the variation of $fCO_2$ with temperature in one seawater sample from the North Atlantic Ocean (strictly speaking, they reported $pCO_2$, but the results apply equally to $fCO_2$). Their measurements indicated that the temperature sensitivity of $fCO_2$ (denoted $v$; Eq. 1) was essentially independent of $fCO_2$ and temperature (i.e., temperature and $\ln(fCO_2)$ have a linear relationship). This apparent near-linearity was consistent with earlier measurements by Kanwisher (1960) on a single seawater sample collected from the vicinity of Woods Hole (Massachusetts,

USA) and with later measurements by Lee and Millero (1995). Takahashi et al. (1993) also claimed that the value of $v$ was no different for higher salinity waters, citing unpublished preliminary data.

However, it has long been known that $v$ is not constant across the ocean. For example, Gordon and Jones (1973) computed $v$ using a finite difference approach across a range of different temperature, salinity, alkalinity and pH values and found that the variability in $v$ could be approximately parameterised as a function of $p\mathrm{CO_2}$. Using a similar approach, but with different parameterisations of the equilibrium constants in their marine carbonate system calculations, Weiss et al. (1982) and Copin-Montegut (1988) each fitted $v$ to functions of $f\mathrm{CO_2}$ and temperature. It has been noted that $v$ may vary with the dissolved inorganic carbon ($T_C$) content of seawater (e.g., Takahashi et al., 2002), or more generally the ratio between $T_C$ and total alkalinity ($A_T$) (e.g., Goyet et al., 1993; Wanninkhof et al., 2022).

There are also several lines of evidence that the relationship between temperature and $\ln(f\mathrm{CO_2})$ is not linear (i.e., $v$ is not constant for a single, isochemical sample). Takahashi et al. (1993) acknowledged this possibility by fitting $\ln(f\mathrm{CO_2})$ to a quadratic function of temperature as well as their linear function, although they did not give any reason to choose one over the other. The fits of Weiss et al. (1982), Copin-Montegut (1988) and Goyet et al. (1993) included various degrees of polynomial temperature dependence. Calculating how $f\mathrm{CO_2}$ changes with temperature using software tools like CO2SYS (Lewis and Wallace, 1998) also shows a non-linear relationship, but its exact form and how much it deviates from linearity both vary widely depending on which parameterisation for the equilibrium constants of carbonic acid is used (McGillis and Wanninkhof, 2006). Data compiled from multiple cruises where $f\mathrm{CO_2}$ was measured at two different temperatures in an equivalent pair of samples, often with accompanying $A_T$ and $T_C$, showed that a linear (constant $v$) adjustment was insufficiently accurate for $\Delta t$ greater in magnitude than about 1 °C (Wanninkhof et al., 2022).

Wanninkhof et al. (2022) recommended calculating $v$ from $A_T$ and $T_C$ in order to make accurate adjustments with greater $\Delta t$, using the carbonic acid dissociation constants of Lueker et al. (2000) and borate:chlorinity of Uppström (1974). However, the requirement for a second marine carbonate system parameter other than $f\mathrm{CO_2}$ is a major disadvantage because, due to ships of opportunity and the variety of (semi-)autonomous sensors available, seawater $f\mathrm{CO_2}$ is by far the most widely measured marine carbonate system parameter in the ocean and it is often the only one available at a particular observation point. A possible workaround would be to use a parameterisation to generate values for a second parameter. For example, across much of the open ocean, $A_T$ can be calculated from temperature and salinity with an uncertainty of less than 10 µmol kg$^{-1}$ (Lee et al., 2006). Under typical conditions, this component alone would propagate through to a less than 0.1 µatm uncertainty in $f\mathrm{CO_2}$ adjusted by 10 °C (calculated with PyCO2SYS; Humphreys et al., 2022). However, this ignores the much larger contribution of uncertainties in the equilibrium constants used in the calculations. These are not well constrained, but using the set suggested by Orr et al. (2018) makes the uncertainty in the same adjusted $f\mathrm{CO_2}$ value several hundred times greater, at around 15 µatm. An adjustment method based on direct measurements of $f\mathrm{CO_2}$ and independent from a second marine carbonate system parameter, avoiding the uncertainties inherent in calculations involving equilibrium constants, is therefore highly desirable.

Given the evidence that the equations of Takahashi et al. (1993) are not sufficiently accurate for the full range of temperature adjustments of $f\mathrm{CO_2}$ required by ocean scientists, why do they remain so widely used? First, they are straightforward to calculate, relying only on variables that will be available with any $f\mathrm{CO_2}$ measurement (i.e., $f\mathrm{CO_2}$ itself and temperature). The more complex existing parameterisations that also fulfil this criterion (Copin-Montegut, 1988; e.g., Gordon and Jones, 1973; Weiss et al., 1982) are not based on up-to-date parameterisations of the carbonic acid dissociation constants, which can lead to differences in adjusted $f\mathrm{CO_2}$ values of over 30 µatm for larger $\Delta t$ (McGillis and Wanninkhof, 2006). But a more fundamental impediment is not knowing what form the relationship between temperature and $\ln(f\mathrm{CO_2})$ should be expected to take from a theoretical perspective. Should it be (roughly) linear, thus deviations from linearity indicate problems with the calculations used to solve the marine carbonate system? Or should it be non-linear, thus indicating some deficiency

with the measurements of Takahashi et al. (1993) – but then which of the non-linear relationships obtained from the different parameterisations of the carbonic acid dissociation constants should be followed?

Here, we aim to provide this missing theoretical basis by developing a new functional form for how $f\text{CO}_2$ and thus $v$ vary with temperature, derived from the equations governing the marine carbonate system and the van 't Hoff equation. We compare this new form with $v$ calculated from a fully resolved marine carbonate system – to which end we have added the calculation of $v$ to PyCO2SYS (Humphreys et al., 2022) – and with laboratory and field data. We compute uncertainties in $v$ and temperature-adjusted $f\text{CO}_2$ values due to experimental uncertainties and variability in $v$ through space and time. We assess the contributions of the different equilibrium constants of the marine carbonate system to $v$, especially those for carbonic acid dissociation, and consider the implications for how these constants are parameterised and how these previously separate aspects can be better integrated in future studies for a more accurate and internally consistent understanding of the marine carbonate system.

## 2 Methods

### 2.1 Adjusting $f\text{CO}_2$ to a different temperature

#### 2.1.1 General theory

The temperature sensitivity of seawater $f\text{CO}_2$ ($v$ in units of % $°\text{C}^{-1}$) is the first derivative of the natural logarithm of $f\text{CO}_2$ with respect to temperature ($t$ in °C) under constant total alkalinity ($A_\text{T}$) and dissolved inorganic carbon ($T_\text{C}$) (see Humphreys et al. (2022) and references therein for more detailed definitions of $A_\text{T}$, $T_\text{C}$ and $f\text{CO}_2$):

$$v = \frac{\partial \ln(f\text{CO}_2)}{\partial t}\bigg|_{A_\text{T},T_\text{C}} \tag{1}$$

noting that here and throughout, $f\text{CO}_2$ must be divided by unit $f\text{CO}_2$ (e.g., 1 µatm if $f\text{CO}_2$ is in µatm) before taking the logarithm, and 'constant $A_\text{T}$ and $T_\text{C}$' implies that all other parameters but temperature are also constant (e.g., salinity and total nutrient contents). We report $v$ in units of % $°\text{C}^{-1}$ (i.e., $10^{-2}$ $°\text{C}^{-1}$).

To adjust an $f\text{CO}_2$ value from one temperature ($t_0$) to another ($t_1$), $v$ must be integrated across the relevant temperature range:

$$\ln[f\text{CO}_2(t_1)] = \ln[f\text{CO}_2(t_0)] + Y \tag{2}$$

where $Y$ denotes the integral

$$Y = \int_{t_0}^{t_1} v \, dt \tag{3}$$

Equivalent to Eq. (2):

$$f\text{CO}_2(t_1) = f\text{CO}_2(t_0) \times \exp(Y) \tag{4}$$

In order to adjust the temperature of a $f\text{CO}_2$ value using Eq. (4), a function for $v$ that can be integrated with respect to temperature is therefore needed.

#### 2.1.2 Experiments to measure $v$

Takahashi et al. (1993) attempted to determine $v$ by measuring how $p\text{CO}_2$ varies with temperature (8 measurements at 7 temperatures from 2.1 to 24.5 °C) in one seawater sample with constant $A_\text{T}$ (reported as "2270 to 2295" µmol kg$^{-1}$) and $T_\text{C}$ (2074 µmol kg$^{-1}$). The practical salinity was 35.380. Phosphate and silicate concentrations were reported as zero while noting the possibility of silicate contamination from the sample bottles. The sample had been part of the laboratory intercomparison exercise reported by Poisson et al. (1990).

We determined a more precise $A_\text{T}$ for the sample by finding the value that, when paired with the $T_\text{C}$ and other parameters above, returned the least-squares best fit to the reported $p\text{CO}_2$ values. When using the Lueker et al. (2000) carbonic acid constants and the borate:chlorinity of Uppström (1974), the $A_\text{T}$ thus obtained with PyCO2SYS (Humphreys et al., 2022)

was 2263 ± 3 µmol kg$^{-1}$, although each set of carbonic acid constants returned its own unique $A_T$. For consistency with modern measurements and the rest of our analysis, we also converted the Takahashi et al. (1993) $pCO_2$ values into $fCO_2$, although this makes no meaningful difference to any of our results.

Takahashi et al. (1993) fitted their $pCO_2$ dataset to linear (subscript $l$) and quadratic (subscript $q$) equations (here written in terms of $fCO_2$):

$$\ln(fCO_2) = b_l t + c_l \tag{5}$$

$$\ln(fCO_2) = a_q t^2 + b_q t + c_q \tag{6}$$

where $a$, $b$ and $c$ are the fitted coefficients. Differentiating with respect to $t$ yields the following for $v$:

$$v_l = b_l \tag{7}$$

$$v_q = 2a_q t + b_q \tag{8}$$

Finally, integrating across the relevant temperature range yields $Y$, following e.g. Takahashi et al. (2009):

$$Y_l = b_l(t_1 - t_0) \tag{9}$$

$$Y_q = a_q(t_1^2 - t_0^2) + b_q(t_1 - t_0) \tag{10}$$

Takahashi et al. (1993) reported values of $b_l = 4.23\%$ °C$^{-1}$, $a_q = -4.35\times10^{-5}$ °C$^{-2}$ and $b_q = 4.33\%$ °C$^{-1}$ with a root-mean-square deviation (RMSD) of 1.0 µatm in $fCO_2$ for both fits. Reproducing these fits with their reported data gives an RMSD of 1.2 µatm for the linear fit and 0.9 µatm for the quadratic.

Lee and Millero (1995) conducted a similar set of measurements as part of a wider experimental study of the marine carbonate system. Again using only one single sample, they measured the variation of $fCO_2$ with temperature at roughly 5 °C
intervals from 5.05 to 35 °C (7 data points). The reproducibility of the $fCO_2$ measurements was reported as ±3 µatm. The properties of the sample were reported as $A_T = 2320.8 \pm 1.6$ µmol kg$^{-1}$, $T_C = 1966.8 \pm 1.2$ µmol kg$^{-1}$, and salinity = 34.95. The experimental setup was a little different, for example with Takahashi et al. (1993) equilibrating the sample with an air headspace that was subsampled for separate measurement and Lee and Millero (1995) continuously circulating the air headspace through the CO$_2$ analyser. Lee and Millero (1995) fitted a linear equation of the form of Eq. (5) to their data, finding
a best-fitting $v_l$ of 4.26% °C$^{-1}$. Refitting their dataset but minimising residuals in $fCO_2$ rather than $\ln(fCO_2)$ returns $b_l = v_l = 4.23\%$ °C$^{-1}$.

## 2.1.3 Approach using van 't Hoff equation

Takahashi et al. (1993) did not give a theoretical basis for either of the forms (linear and quadratic; Eqs. 5 and 6) that they fitted to their dataset nor did they give any reason to choose one over the other. But, as will be discussed later, the form chosen
has an important effect on adjustments, especially over greater $\Delta t$ (Sect. 3.1), and on uncertainty propagation (Sect. 3.2). Here, we provide the theoretical basis for an alternative form based on the equations governing the marine carbonate system and the van 't Hoff equation, which is derived from the well-known thermodynamic equation for the Gibbs free energy of chemical reactions.

Seawater $fCO_2$ is related to $A_T$ and $T_C$ by (e.g., Humphreys et al., 2018)

$$fCO_2 = \frac{K_2^*}{K_0' K_1^*} \frac{[HCO_3^-]^2}{[CO_3^{2-}]} \tag{11}$$

The asterisks in $K_1^*$ and $K_2^*$ show that they are stoichiometric equilibrium constants, based on the substance contents of the equilibrating species, rather than thermodynamic constants based on their activities; the prime symbol in $K_0'$ shows that it is a hybrid equilibrium constant, based on the substance content of CO$_2$(aq) and the fugacity of CO$_2$(g) (Zeebe and Wolf-Gladrow, 2001). Following Humphreys et al. (2018), $A_T$ and $T_C$ can be approximated with $A_x$ and $T_x$, which include only the bicarbonate
(HCO$_3^-$) and carbonate (CO$_3^{2-}$) terms:

$$A_T \approx A_x = [HCO_3^-] + 2[CO_3^{2-}] \tag{12}$$

$$T_C \approx T_x = [HCO_3^-] + [CO_3^{2-}] \tag{13}$$

In typical seawater, the approximations in Eqs. (12) and (13) are more than 99% accurate for $T_C$ and more than 97% accurate for $A_T$. These approximations have previously been used to understand how $f\mathrm{CO}_2$ responds to changes in $A_T$ and $T_C$ under constant temperature (Humphreys et al., 2018). Equations (12) and (13) can be rearranged for $[\mathrm{HCO}_3^-]$ and $[\mathrm{CO}_3^{2-}]$:

$$[\mathrm{HCO}_3^-] = 2T_x - A_x \tag{14}$$

$$[\mathrm{CO}_3^{2-}] = A_x - T_x \tag{15}$$

Under this approximation, both $[\mathrm{HCO}_3^-]$ and $[\mathrm{CO}_3^{2-}]$ are thus constant under constant $A_x$ and $T_x$, so temperature affects $f\mathrm{CO}_2$ only through the $K_2^*/K_0'K_1^*$ term in Eq. (11). To relate this to $v$, we need to quantify this sensitivity to temperature:

$$\frac{\mathrm{d}\ln(f\mathrm{CO}_2)}{\mathrm{d}t} \approx \frac{\mathrm{d}}{\mathrm{d}t}\ln\left(\frac{K_2^*}{K_0'K_1^*}\right) = \frac{\mathrm{d}\ln K_2^*}{\mathrm{d}t} - \frac{\mathrm{d}\ln K_0'}{\mathrm{d}t} - \frac{\mathrm{d}\ln K_1^*}{\mathrm{d}t} \tag{16}$$

The van 't Hoff equation provides a theoretical basis for the first-order response of equilibrium constants to temperature:

$$\frac{\mathrm{d}\ln K}{\mathrm{d}t} = \frac{\Delta_r H^\ominus}{Rt_K^2} \tag{17}$$

where $\Delta_r H^\ominus$ is the standard enthalpy of reaction, $R$ the ideal gas constant and $t_K$ temperature in K. Substituting Eq. (17) into Eq. (16) gives

$$\frac{\mathrm{d}\ln(f\mathrm{CO}_2)}{\mathrm{d}t} \approx v_x = \frac{\Delta_r H_2^\ominus - \Delta_r H_0^\ominus - \Delta_r H_1^\ominus}{Rt_K^2} \tag{18}$$

Equation (18) shows that, if the approximations $A_x$ and $T_x$ are sufficiently accurate, then to first order, $\ln(f\mathrm{CO}_2)$ should therefore be proportional to $1/t_K$, rather than to $t$.

Based on reported standard enthalpies of formation at 298.15 K (Ruscic and Bross, 2023), and assuming that these are constant under the temperature range of interest, the numerator in Eq. (18) is $25288 \pm 98$ J mol$^{-1}$ (Appendix A). Given the approximations involved, Eq. (18) is unlikely to provide correct values for $v$ in seawater directly by using this value, but it does suggest a revised form that could be fitted to experimental data such as that of Takahashi et al. (1993):

$$\ln(f\mathrm{CO}_2) = c_h - \frac{b_h}{Rt_K} \tag{19}$$

Following the same steps taken in Sect. 2.1.2 for the linear and quadratic forms of Takahashi et al. (1993), we find the following:

$$v_h = \frac{b_h}{Rt_K^2} \tag{20}$$

$$\Upsilon_h = \frac{b_h}{R}\left(\frac{1}{t_{K,0}} - \frac{1}{t_{K,1}}\right) \tag{21}$$

We fitted Eq. (19) to the Takahashi et al. (1993) dataset, optimising residuals in $f\mathrm{CO}_2$ rather than $\ln(f\mathrm{CO}_2)$ to avoid implicitly weighting the lower-$f\mathrm{CO}_2$ values more heavily. The least-squares best fit $b_h$ was $28995 \pm 235$ J mol$^{-1}$, with an RMSD of 2.9 µatm in $f\mathrm{CO}_2$. Fitting Eq. (19) in the same way to the Lee and Millero (1995) dataset gave a least-squares best fit $b_h$ of $30794 \pm 357$ J mol$^{-1}$, with an RMSD of 2.8 µatm. The different $b_h$ values considered here are summarised in Table 1, including the parameterisation developed later in Sect. 2.4, which gives a range of possible values.

**Table 1: Overview of the different values for $b_h$ (Eqs. 19-21) that are discussed in this article.**

| $b_h$ / J mol$^{-1}$ | Corresponding symbol for $v$ | Description |
|---|---|---|
| $25288 \pm 98$ | $v_x$ | Theoretical value based on standard enthalpies of formation |
| $28995 \pm 235$ | $v_h^{\mathrm{Ta93}}$ | Fitted to Takahashi et al. (1993) dataset |
| $30794 \pm 357$ | $v_h^{\mathrm{LM95}}$ | Fitted to Lee and Millero (1995) dataset |
| c. $25000 - 30700$ | $v_p$ | Range of values across the global surface ocean in the monthly mean OceanSODA-ETZH data product |

## 2.2 Uncertainty in the experiment

### 2.2.1 Propagation equations

To quantify uncertainties in $f\mathrm{CO_2}$ resulting from temperature adjustments one needs to know the uncertainty in $v$ and $Y$ for whichever form is being used. Here, we derive uncertainty propagation equations following JCGM (2008) for the linear (Eq. 5), quadratic (Eq. 6) and van 't Hoff (Eq. 19) forms.

For the linear and van 't Hoff forms, uncertainty propagation is straightforward:

$$\sigma^2(v_l) = \sigma^2(b_l) \tag{22}$$

$$\sigma^2(Y_l) = \sigma^2(b_l) \times (t_1 - t_0)^2 \tag{23}$$

$$\sigma^2(v_h) = \frac{\sigma^2(b_h)}{\left(Rt_\mathrm{K}^2\right)^2} \tag{24}$$

$$\sigma^2(Y_h) = \frac{\sigma^2(b_h)}{R^2}\left(\frac{1}{t_{\mathrm{K},0}} - \frac{1}{t_{\mathrm{K},1}}\right)^2 \tag{25}$$

where $\sigma^2$ is uncertainty expressed as a variance.

For the quadratic form, covariances between the fitted coefficients must be included. In general:

$$\sigma^2(p) = \mathbf{J}(p) \cdot \mathbf{U}_q \cdot \mathbf{J}(p)^\mathrm{T} \tag{26}$$

where $p$ is the parameter of interest and the uncertainty matrix for the coefficients is

$$\mathbf{U}_q = \begin{bmatrix} \sigma^2(a_q) & \sigma(a_q, b_q) \\ \sigma(a_q, b_q) & \sigma^2(b_q) \end{bmatrix} \tag{27}$$

The Jacobian matrices for $v_q$ and $Y_q$ are

$$\mathbf{J}(v_q) = \begin{bmatrix} \frac{\partial v_q}{\partial a_q} & \frac{\partial v_q}{\partial b_q} \\ \vdots & \vdots \end{bmatrix} = \begin{bmatrix} 2t & 1 \\ \vdots & \vdots \end{bmatrix} \tag{28}$$

$$\mathbf{J}(Y_q) = \begin{bmatrix} \frac{\partial Y_q}{\partial a_q} & \frac{\partial Y_q}{\partial b_q} \\ \vdots & \vdots \end{bmatrix} = \begin{bmatrix} (t_1^2 - t_0^2) & (t_1 - t_0) \\ \vdots & \vdots \end{bmatrix} \tag{29}$$

Thus for a single measurement, Eq. (26) gives

$$\sigma^2(v_q) = 4t^2\sigma^2(a_q) + \sigma^2(b_q) + 4t\sigma(a_q, b_q) \tag{30}$$

$$\sigma^2(Y_q) = (t_1^2 - t_0^2)^2\sigma^2(a_q) + (t_1 - t_0)^2\sigma^2(b_q) + 2(t_1^2 - t_0^2)(t_1 - t_0)\sigma(a_q, b_q) \tag{31}$$

Finally, uncertainty in $Y$ can be converted into uncertainty in $\exp(Y)$ for use in Eq. (4) with

$$\sigma^2[\exp(Y)] = [\exp(Y)]^2 \times \sigma^2(Y) \tag{32}$$

### 2.2.2 Uncertainties in fitted parameters

To use the equations in Sect. 2.2.1, the uncertainties in the fitted parameters from Eqs. (5), (6) and (19) must be known. We computed these by running Monte-Carlo simulations of the Takahashi et al. (1993) experiment, as this dataset is the most widely used in practice.

We used the Takahashi et al. (1993) temperature and $f\mathrm{CO_2}$ measurements as the starting point for the simulations. For each iteration of the simulation (total iterations = $10^6$), we generated a random number for each of temperature and $f\mathrm{CO_2}$, which were added uniformly across the entire dataset to represent systematic errors (trueness), and then a different random number for each temperature and $f\mathrm{CO_2}$ data point, to represent random errors in each measurement (precision). The distributions used for the random numbers are shown in Table 2. The simulated temperature and $f\mathrm{CO_2}$ values (starting point plus systematic and random errors) were then fitted to Eqs. (5), (6) and (19) separately in each iteration. All the forms of $v$ and $Y$ were also computed based on the fit at each iteration.

The variances $\sigma^2(b_l)$ and $\sigma^2(b_h)$ and the variance-covariance matrix $\mathbf{U}_q$ (Eq. 27) were computed from the fitted coefficients across the full set of iterations.

### 2.2.3 Validation of uncertainty propagation

The uncertainty propagation equations in Sect. 2.2.1 have all been validated by two comparisons: first, with the variances of the $v$ and $Y$ values computed across all the simulations in Sect. 2.2.2, and second, by computing the derivatives required for propagation using automatic differentiation (Maclaurin, 2016) rather than manually. The values from the equations agreed to within 0.01% of those based on the simulations and to within $10^{-12}$% with the calculations based on automatic differentiation. The former difference is greater (albeit still negligible) because of non-linearities in the forms of $v$ that are not accounted for in the first-order derivatives used for uncertainty propagation here. The latter difference is attributable to rounding errors at the limit of computer precision.

**Table 2. Uncertainties in temperature and seawater $f\mathrm{CO_2}$ assumed for our Monte-Carlo simulations of the Takahashi et al. (1993) experiment.**

| Parameter | Type | Distribution | Spread* | Source / explanation |
|---|---|---|---|---|
| Temperature | Random | Uniform | 0.01 °C | Temperature reported to two decimal places |
| | Systematic | Normal | 0.005 °C | WOCE-era accuracy for oceanographic temperature measurements |
| $f\mathrm{CO_2}$ | Random | Normal | 2 µatm | Paper states this as uncertainty; unclear if referring to accuracy or precision |
| | Systematic | Normal | 2 µatm | As above |

*Spread indicates the total range for uniform distributions or the standard deviation for normal distributions, with both distributions centred at zero.

## 2.3 PyCO2SYS extension and calculations

PyCO2SYS is a free and open source Python package which can be used to solve the marine carbonate system, that is, to calculate the equilibrium balance of the main acid-base systems in seawater and related properties (Humphreys et al., 2022). PyCO2SYS was originally based on the MATLAB/GNU Octave program CO2SYS.m (v2.0.5; Orr et al., 2018), itself part of a family of similar software tools beginning with the original CO2SYS for MS-DOS (Lewis and Wallace, 1998).

### 2.3.1 Calculating $v$ in PyCO2SYS

In PyCO2SYS, $v$ can be calculated by finite differences or by automatic differentiation (Maclaurin, 2016). The accuracy of the former depends on the size of the difference used, whereas the latter gives the exact result directly (Humphreys et al., 2022). For consistency with the chemical buffer factors in PyCO2SYS, we used automatic differentiation to calculate $v$. We constructed a function that evaluates $\ln(f\mathrm{CO_2})$ from $A_T$, $T_C$, $t$ and all other relevant parameters (salinity, pressure, nutrients) and applied automatic differentiation to determine its derivative with respect to temperature. The function runs after $A_T$ and $T_C$ have been determined from whichever pair of known parameters the user provides. This was added to the set of results returned by PyCO2SYS as of v1.8.3 (Humphreys et al., 2024).

Previous versions of PyCO2SYS could already calculate $\partial f\mathrm{CO_2}/\partial t$ using forward finite difference derivatives as part of the uncertainty propagation tool. $v$ can be calculated from $\partial f\mathrm{CO_2}/\partial t$:

$$v = \frac{\partial f\mathrm{CO_2}}{\partial t} \, (f\mathrm{CO_2})^{-1} \tag{33}$$

We used the finite-difference $\partial f\mathrm{CO_2}/\partial t$ and Eq. (33) to validate the values from automatic differentiation. Across $10^4$ random combinations of $A_T$, $T_C$, salinity, $t$, pressure, nutrient contents, and carbonic acid dissociation constant parameterisation, we found that the two approaches were consistent, with a mean absolute difference less than $10^{-4}$% °C$^{-1}$ (or 0.01%) and a maximum absolute difference less than 0.1% °C$^{-1}$. This check has been incorporated into the PyCO2SYS test suite to ensure continued consistency in future versions.

### 2.3.2 Adjusting $f\mathrm{CO_2}$ to different temperatures

Prior to v1.8.3, PyCO2SYS could already convert $f\mathrm{CO_2}$ (also $p\mathrm{CO_2}$, $[\mathrm{CO_2(aq)}]$ or $x\mathrm{CO_2}$) values from one temperature to another when no second carbonate system variable was provided, but using only Eq. (10), the quadratic fit of Takahashi et al. (1993). In v1.8.3, we have added the option to use the other methods discussed here instead (i.e., the linear fit of Takahashi et al. (1993) and the van 't Hoff form with several different options for $b_h$, including its parameterisation in Sect. 2.4).

Like in previous versions of PyCO2SYS, the same adjustment is applied if any of $p\mathrm{CO_2}$, $[\mathrm{CO_2(aq)}]$ or $x\mathrm{CO_2}$ are given as the single known marine carbonate system parameter, as these can all be inter-converted with each other and with $f\mathrm{CO_2}$ without knowledge of a second marine carbonate system parameter (Humphreys et al., 2022).

### 2.3.3 Components of $v$

The effect of temperature on $f\mathrm{CO_2}$ propagates through from all the different equilibrium constants of the marine carbonate system. The total $v$ is the sum of the partial derivatives with respect to each constant:

$$v = \sum_K \left( \frac{\partial \ln(f\mathrm{CO_2})}{\partial K} \cdot \frac{\partial K}{\partial t} \right) \tag{34}$$

where $K$ is the set of equilibrium constants and other properties that are sensitive to $t$ and involved in calculating $f\mathrm{CO_2}$ from $A_\mathrm{T}$ and $T_\mathrm{C}$, specifically, the $\mathrm{CO_2}$ solubility constant $K_0'$, the carbonic acid dissociation constants $K_1^*$ and $K_2^*$, the borate equilibrium constant $K_B^*$, and the water dissociation constant $K_w^*$. The dissociation constants for phosphoric acid, orthosilicic acid, hydrogen sulfide, ammonia, sulfate and hydrogen fluoride also play a role, but their contributions to $v$ are negligible, being much less than 1% in the modern surface ocean, so we did not include them further in our analysis. (Their effects are still included in the total $v$ calculated with automatic differentiation, described in Sect. 2.3.1.)

We used the forward finite difference derivatives calculated by PyCO2SYS (Humphreys et al., 2022) to evaluate $\partial f\mathrm{CO_2}/\partial K$ and $\partial K/\partial t$ and combined these with Eq. (33) to quantify the influence of each component on $v$.

### 2.3.4 Additional carbonic acid parameterisation

We incorporated the parameterisation of Papadimitriou et al. (2018) for the carbonic acid dissociation constants into PyCO2SYS. This parameterisation covers a range of temperature (–6 to 25 °C) and salinity (33 to 100) conditions appropriate for sea-ice brines; previously, the lowest valid temperature for any parameterisation in PyCO2SYS was –1.7 °C and the highest salinity was 50. The calculations were tested against the check values provided by Papadimitriou et al. (2018) in their Table 3 and these tests were incorporated into the PyCO2SYS automatic test suite (Humphreys et al., 2022).

### 2.4 Variability in $v$ and $b_h$

We used the OceanSODA-ETZH data product (Gregor and Gruber, 2021; dataset version 5, downloaded 4 December 2023) to investigate spatiotemporal variability in surface ocean $v$. OceanSODA-ETZH provides fields of several marine carbonate system and other hydrographic parameters gridded across the surface ocean (1° × 1°) and through time (monthly from 1985 to 2018). The parameter fields were generated using an ensemble cluster-regression approach based on observations in SOCAT (Bakker et al., 2016) and GLODAP (Lauvset et al., 2016). Other similar data products exist, but the main patterns and variability in surface ocean carbonate chemistry are well enough constrained that the choice of a particular data product will not significantly affect the global-scale and time-averaged analyses conducted here.

First, at each grid point and for each month, we computed the mean across all years of temperature, salinity, $A_\mathrm{T}$ and $T_\mathrm{C}$. We then computed $v$ from these monthly mean gridded fields using PyCO2SYS v1.8.3 (Sect. 2.3), with the carbonic acid dissociation constants of Lueker et al. (2000), the bisulfate dissociation constant of Dickson (1990a), the hydrogen fluoride dissociation constant of Dickson and Riley (1979) and the total borate to chlorinity ratio of Uppström (1974). OceanSODA-

ETZH does not include nutrient fields, so their concentrations were set to zero, but we do not expect this to affect the results because, as noted in Sect. 2.3.2, their effects on $\upsilon$ are negligible.

We also used the OceanSODA-ETZH data product to parameterise $b_h$. We calculated $f\mathrm{CO_2}$ at 50 evenly spaced temperatures from –1.8 to 35.83 °C (i.e., the full range in the data product) at each monthly mean grid point. This simulates running an experiment like that of Takahashi et al. (1993) at each point. Next, we fit Eq. (19) to the generated $t$ and $f\mathrm{CO_2}$ data to find the best fitting $b_h$ value at each point. Finally, we collated all the $b_h$ values and fitted them using a function of the form

$$b_h = u_0 + u_1 t + u_2 s + u_3 f + u_4 t^2 + u_5 s^2 + u_6 f^2 + u_7 ts + u_8 tf + u_9 sf \tag{35}$$

where $s$ is practical salinity, $f$ is $f\mathrm{CO_2}$ and $u_0$-$u_9$ are the fitted coefficients. The coefficients and their variance-covariance matrix are provided in the Supplementary Information (Supp. Tables 1-2). Values of $\upsilon$ computed using Eq. (35) for $b_h$ are denoted $\upsilon_p$.

Fitting the parameterisation to a data product has an advantage over using regular grids of the predictors in that it guarantees including all combinations of predictors that do occur in the real ocean and avoids combinations that do not. It also causes the fit to be weighted more strongly by more prevalent combinations of conditions, leading to improved performance in global applications. However, some regions are less well represented by this global fit (Supp. Fig. 1), so regionally specialised fits might work better for some studies.

A difficulty with applying our parameterisation of $b_h$ is that in-situ $f\mathrm{CO_2}$ is one of the predictors. This may not always be known (e.g., if discrete measurements were conducted at a fixed temperature in the laboratory), and using a different $f\mathrm{CO_2}$ will not yield an accurate $b_h$. To mitigate this issue, if in-situ $f\mathrm{CO_2}$ is not known before the conversion, we use the constant $b_h$ fitted to the Takahashi et al. (1993) dataset (Sect. 2.1.3) to preliminarily adjust $f\mathrm{CO_2}$ to the in situ temperature, and then use this preliminary $f\mathrm{CO_2}$ to compute $b_h$ with Eq. (35), before recalculating in situ $f\mathrm{CO_2}$ with the computed $b_h$. This approach is incorporated into the PyCO2SYS implementation, with the user required to identify whether the input- or output-condition $f\mathrm{CO_2}$ values represent in-situ conditions for this purpose.

## 2.5 Open source software

All of the analysis presented here was conducted using the free and open source Python programming language (Python Software Foundation; https://www.python.org/). The main packages used, which are also free and open source, include NumPy (Harris et al., 2020), Autograd (Maclaurin, 2016) and xarray (Hoyer and Hamman, 2017) for data manipulation and general calculations; SciPy (Virtanen et al., 2020) for curve fitting; Matplotlib (Hunter, 2007) and Cartopy (Met Office, 2010) for data visualisation; and PyCO2SYS (Humphreys et al., 2022) for marine carbonate system calculations. Geographical features on maps were drawn using data that are freely available from Natural Earth (https://www.naturalearthdata.com/).

## 3 Results and discussion

### 3.1 Functional form of $\upsilon$

Our first goal is to determine whether the newly proposed form $\upsilon_h$ (Eq. 20) can accurately represent the temperature-sensitivity of seawater $f\mathrm{CO_2}$. To this end, we assess the accuracy of the approximations $A_x$ and $T_x$ made in developing Eq. (20) by comparing $\upsilon_h$ with calculations from a fully resolved marine carbonate system (Sect. 3.1.1). Next, we examine how well the different approaches to calculating $\upsilon$ agree with direct measurements, first for laboratory experiments with measurements of $f\mathrm{CO_2}$ at multiple temperatures for a single seawater sample (Sect. 3.1.2), and then across a global dataset with pairs of measurements at two different temperatures (Sect. 3.1.3).

### 3.1.1 Accuracy of the approximations

We made several approximations to develop Eq. (20) (Sect. 2.1.3). First, all non-carbonate components of $A_T$ (Eq. 12) and the contribution of $[\mathrm{CO_2(aq)}]$ to $T_C$ (Eq. 13) were neglected. Second, the simple integrated van 't Hoff equation (Eq. 17) was

assumed to accurately represent how the various equilibrium constants vary with temperature. Third, when calculating $b_h$ from standard enthalpies of formation (Appendix A), these were assumed to be constant across the relevant temperature range. We must therefore check whether Eq. (20) can still capture the sensitivity of $f\mathrm{CO_2}$ to temperature accurately enough for our purposes before applying it to oceanographic data.

The first approximation above was also used to investigate the complementary question of how $f\mathrm{CO_2}$ varies with $A_T$ and $T_C$ under constant temperature (Humphreys et al., 2018). There, it was observed that the gradient of isocaps (i.e., lines of constant $f\mathrm{CO_2}$ through $A_T$-$T_C$ phase-space) was correlated with $f\mathrm{CO_2}$. The approximations $A_x$ and $T_x$ were used as the basis for a simplified representation of the marine carbonate system from which equations were derived to explain the observed correlation. This successful application to a closely related problem provides a first indication that the same approximations may also be suitable here.

We can do a more direct test using software like PyCO2SYS, which does not make any of the approximations above when calculating how $f\mathrm{CO_2}$ varies with temperature under constant $A_T$ and $T_C$ (Humphreys et al., 2022). Instead, the complete equations for $A_T$ and $T_C$ are used, and the equilibrium constants are parameterised with multiple temperature-, salinity- and pressure-sensitive terms, which implicitly include variations in enthalpies of formation. Such calculations can therefore be used to quantify the effect of the approximations on the accuracy of Eq. (20). Equation (20) cannot be fitted directly to $f\mathrm{CO_2}$ data because $v$ represents the derivative with respect to temperature; instead, we need to fit Eq. (19) to obtain the unknown $b_h$ which can then be used in Eq. (20) for $v_h$. If we use PyCO2SYS to calculate $f\mathrm{CO_2}$ across a range of temperatures with constant $A_T$ and $T_C$, denoted $f\mathrm{CO_2}(A_T, T_C)$, and then fit Eq. (19) to the results (Sect. 2.4), denoted $f\mathrm{CO_2}(v_p)$, then the residuals from the fit should reveal any reduction in accuracy due to the approximations. Large residuals would suggest that at least one approximation was unsuitable for modelling $v$, while small residuals would mean that the approximated system does still capture the main features of the $t$-$f\mathrm{CO_2}$ relationship that are needed to model $v$.

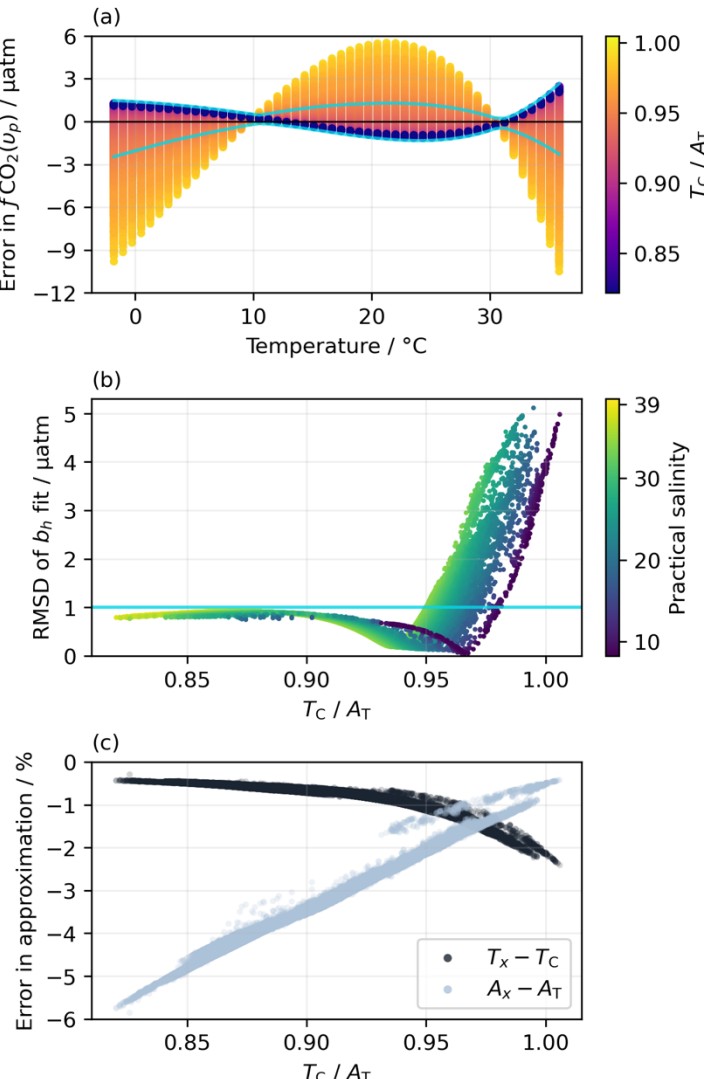

**Figure 1: (a) Residuals in $f\text{CO}_2$ for the best fit of Eq. (19) to the $f\text{CO}_2$ values calculated from a fully solved marine carbonate system**
**at each monthly mean grid point in the OceanSODA-ETZH data product (Sect. 2.4), coloured by $T_C/A_T$. Each point in OceanSODA-ETZH thus appears as a line of points across (a) with constant $T_C/A_T$. (b) Variation of the RMSD of the residuals shown in (a) with $T_C/A_T$, coloured by practical salinity. Each point in OceanSODA-ETZH thus appears as a single point in (b), which shows the RMSD of the corresponding line of points in (a). The blue lines on (a) and (b) correspond to an RMSD of 1 µatm: points with RMSD less than 1 µatm fall between the blue lines in (a) and below the horizontal blue in (b). Less than 3% of the points have an RMSD greater**
**than 1 µatm. (c) Variation of the percentage errors in the approximations $A_x$ (Eq. 12) and $T_x$ (Eq. 13) with $T_C/A_T$ at each monthly mean grid point in OceanSODA-ETZH.**

For over 97% of the monthly mean grid points in OceanSODA-ETZH, the RMSD of the residuals between $f\text{CO}_2$ calculated from Eq. (19) and from $A_T$ and $T_C$ is less than 1 µatm, with maximum deviations of ±2 µatm (Fig. 1). At lower $T_C/A_T$ (< ~0.95), $f\text{CO}_2(v_p)$ is generally greater than $f\text{CO}_2(A_T, T_C)$ at low (< 10 °C) and high (> 30 °C) temperatures but less than $f\text{CO}_2(A_T, T_C)$
at intermediate temperatures (10-30 °C), but this switches to the opposite pattern at higher $T_C/A_T$ (Fig. 1a). We could not find a relationship between the switchover point, which has a very low RMSD, and some property of chemical buffering in seawater, although the $T_C/A_T$ value at which it occurs is correlated with salinity (Fig. 1b).

       As context for interpreting this < 1 µatm RMSD, the uncertainty in $f\text{CO}_2$ calculated from $A_T$ and $T_C$ that propagates from 0.1% uncertainties in each of $A_T$ and $T_C$ (i.e., the GOA-ON climate quality goal; Newton et al., 2015) varies from 2 µatm
at –1 °C to 9 µatm at 35 °C. The uncertainties in the equilibrium constants involved in this calculation are poorly known, but using the set proposed by Orr et al. (2018) roughly doubles the total uncertainty in calculated $f\text{CO}_2$ given above. The < 1 µatm RMSD between $f\text{CO}_2(v_p)$ and $f\text{CO}_2(A_T, T_C)$ found for 97% of the surface ocean is thus much smaller than the uncertainties in the marine carbonate system calculations themselves. It is also very similar to the RMSD of the linear and quadratic fits to the Takahashi et al. (1993) dataset (Sect. 2.1.2) and better than the GOA-ON climate quality goal for $f\text{CO}_2$ measurements.

In the remaining < 3% of the global surface ocean, the RMSD can increase to over 5 μatm, which is less satisfactory, but still within the GOA-ON weather quality goal (Newton et al., 2015). These higher RMSD values are constrained to $T_C/A_T$ values above 0.95 and to certain regions and seasons, primarily the Baltic Sea during winter and the near-coastal Arctic Ocean (Supp. Fig. 1). In these regions, the RMSD is correlated with several marine carbonate system properties, but one of the strongest correlations is with $T_C/A_T$ (Fig. 1b). The cause of the increased RMSD is likely inaccuracy in the approximation $T_x$ (Eq. 13). The fraction of $T_C$ comprised of $[CO_2(aq)]$, which is ignored in $T_x$, increases with $T_C/A_T$ (Fig. 1c). Consequently, the approximation that $[HCO_3^-]^2/[CO_3^{2-}]$ is constant across different temperatures (Eq. 16), which emerges from the definitions of $A_x$ and $T_x$ (Eqs. 12-15), becomes less accurate with increasing $T_C/A_T$. The fraction of non-carbonate alkalinity, which is ignored in $A_x$, decreases with increasing $T_C/A_T$, which makes $A_x$ more accurate at higher $T_C/A_T$ (Fig. 1c), so $A_x$ should not be the cause of the increased RMSD. The RMSD of the $b_h$ fit is positively correlated with the difference $T_C - T_x$, but negatively correlated with $A_T - A_x$ (Supp. Fig. 3), supporting the hypothesis that inaccuracy in $T_x$ rather than $A_x$ drives the increased RMSD observed at higher $T_C/A_T$.

The ~59% of data points with $T_C/A_T < 0.9$ (in dark purple on Fig. 1a) all have the same pattern of residuals with an RMSD that is virtually independent of $T_C/A_T$ and less than 1 μatm. Their independence from $T_C/A_T$ suggest that these residuals result from the additional temperature-dependent terms beyond $1/t_K$ from the van 't Hoff equation that are included in parameterisations of the equilibrium constants (e.g., Weiss, 1974; Dickson, 1990b; Lueker et al., 2000). Including such additional terms in Eq. (19) (and thus also in Eqs. 20-21) might allow it to better capture the pattern of $\ln(f CO_2)$ variation with $t$, reducing the RMSD. Were $v_h$ to be adapted to include additional fitted parameters, the functional forms included should have a theoretical basis (e.g., Clarke and Glew, 1966) rather than being purely empirical, to ensure accurate uncertainty propagation (Sect. 3.2).

However, to include additional fitted parameters now would be premature. There are major differences between the different carbonic acid parameterisations (e.g., Supp. Fig. 2) that are not yet fully understood, and Fig. 1a-b would show different patterns if reproduced using a different parameterisation. Even though it is the parameterisation given by the best practice guide (Dickson et al., 2007), it is still well-known that Lueker et al. (2000) does not perfectly represent the marine carbonate system to the accuracy with which we can measure it (e.g., Álvarez et al., 2020; Carter et al., 2024; Wang et al., 2023; Woosley, 2021; Woosley and Moon, 2023), and there are outstanding issues to resolve with for example the contribution of organic matter to total alkalinity (e.g., Ulfsbo et al., 2015; Hu, 2020; Kerr et al., 2021; Lee et al., 2024) and the borate:chlorinity ratio (e.g., Lee et al., 2010; Fong and Dickson, 2019). Furthermore, there are no robust uncertainty estimates for the equilibrium constants that can be propagated through to determine a meaningful uncertainty in $v_{Lu00}$ (Orr et al., 2018). The carbonic acid and other equilibrium constant parameterisations are not optimised for this derivative, emergent property ($v$) and direct measurements like those of Takahashi et al. (1993), with which more subtle variations about the first-order form in Eq. (20) can be verified, are scarce (Sect. 3.1.2).

Even with these issues remaining unresolved, we conclude that for over 97% of the global surface ocean, Eq. (20) with all its approximations (Sect. 2.1.3) can capture the pattern of $f CO_2$ sensitivity to temperature according to calculations from $A_T$ and $T_C$ with the carbonic acid dissociation constants of Lueker et al. (2000) to within an uncertainty of 1 μatm.

### 3.1.2 Direct measurements of $v$

Having established how well $v_h$ from Eq. (20) agrees with calculations from a fully solved marine carbonate system, we next consider how these and other approaches compare with the scarce experimental evidence available.

The linear ($v_l$) and quadratic ($v_q$) forms of Takahashi et al. (1993) both agree well with their own measurements, so it is not clear which (if either) is more appropriate. The RMSD for $v_q$ (0.9 μatm) is lower than that for $v_l$ (1.2 μatm), but this reduction is inevitable because $v_q$ has an extra adjustable coefficient (Eqs. 5-8). There is a meaningful difference between $v_l$

and $v_q$ only above ~25 °C (Fig. 2a), where $v_q$ is lower, meaning that $f\mathrm{CO_2}$ would increase less with warming than suggested by $v_l$. For example, adjusting from 25 to 35 °C would give an $f\mathrm{CO_2}$ about 17 µatm lower with $v_q$ than with $v_l$.

Next, we consider the form proposed here in Eq. (19). When $b_h$ is fixed to the value based on standard enthalpies of reaction (Sect. 2.1.3), so only the intercept ($c_h$) can be optimised, the resulting curve $v_x$ has a rather different pattern that is inconsistent with the measurements (Fig. 2a). However, also allowing $b_h$ to be fitted to the measurements of Takahashi et al. (1993) ($v_h^{\mathrm{Ta93}}$) gives a pattern similar to $v_q$, but with stronger curvature. Although it agrees with the Takahashi et al. (1993) measurements less well than $v_l$ and $v_q$ do, with an RMSD of 2.9 µatm as opposed to ~1 µatm, the $v_h$ line still falls within their uncertainty window. Like for $v_q$, the most significant deviation from $v_l$ occurs above 25 °C, where Takahashi et al. (1993) did not report any measurements. Adjusting from 25 to 35 °C would give and $f\mathrm{CO_2}$ about 45 µatm lower with $v_h^{\mathrm{Ta93}}$ than with $v_l$.

There are 18 different parameterisations of the carbonic acid dissociation constants in PyCO2SYS v1.8.3 and each gives a different pattern for $v$. However, they are mostly negatively correlated with temperature, like $v_q$ and $v_h^{\mathrm{Ta93}}$, so they all give lower $f\mathrm{CO_2}$ at higher temperatures than $v_l$ does (Supp. Fig. 2). Two of the parameterisations are highlighted here.

The parameterisation of Millero (1979) ($v_{\mathrm{Mi79}}$) agrees remarkably well with $v_x$ (Fig. 2). This parameterisation was designed for zero-salinity freshwater and it excludes many of the non-carbonate equilibria that are active in seawater, such as that of borate.

The parameterisations that are (at least partly) based on the measurements of Mehrbach et al. (1973) fit the Takahashi et al. (1993) dataset better than those that do not, with the Lueker et al. (2000) parameterisation ($v_{\mathrm{Lu00}}$) having one of the best fits (RMSD = 2.6 µatm) and falling within the uncertainty of the measurements (Fig. 2a). The pattern of $v_{\mathrm{Lu00}}$ is very similar to $v_h^{\mathrm{Ta93}}$ (Fig. 2b). The similarity was expected, given the results discussed in Sect. 3.1.1 and the $T_C/A_T$ of ~0.92 for the sample used by Takahashi et al. (1993). But the similarity above 25 °C is more remarkable in this case, because in Sect. 3.1.1, the $v_h$ curves were fitted to data spanning the full temperature range from –1.8 to 35.83 °C, whereas here, there were no data above 24.5 °C (nor below 2.1 °C) to guide the shape of the fitted $v_h^{\mathrm{Ta93}}$ curve. Thus even without having a wide enough temperature range for the data to show convincing curvature in $v$, the extrapolation of Eq. (20) still matches the calculations with PyCO2SYS very well, unlike $v_l$ and $v_q$.

There are two main reasons for the differences between $v_x$ and $v_h^{\mathrm{Ta93}}$, or equivalently, between $v_{\mathrm{Mi79}}$ and $v_{\mathrm{Lu00}}$. First, $v_h^{\mathrm{Ta93}}$ and $v_{\mathrm{Lu00}}$ include additional equilibria that are not present in $v_x$ and $v_{\mathrm{Mi79}}$. Some of these extra equilibria are explicitly part of the alkalinity equation in PyCO2SYS (e.g., borate). Others are implicit, such as the formation of $\mathrm{CaCO_3^0}$ and $\mathrm{MgCO_3^0}$ ion pairs (Millero and Pierrot, 1998), which are incorporated within $[\mathrm{CO_3^{2-}}]$ and $K_2^*$ in PyCO2SYS (Humphreys et al., 2022). The approximation that all non-carbonate species are negligible within $A_T$ (Eq. 12) is thus more accurate for $v_x$ and $v_{\mathrm{Mi79}}$ (hence their mutual agreement) than it is for $v_h^{\mathrm{Ta93}}$ and $v_{\mathrm{Lu00}}$. Second, the van 't Hoff equation with standard enthalpies of reaction predicts the temperature sensitivity of thermodynamic equilibrium constants. However, chemical equilibria in seawater are modelled using stoichiometric equilibrium constants, which have an additional temperature dependency related to the major ion composition and ionic strength of the solution (Pitzer, 1991). Both reasons would affect $v_h^{\mathrm{Ta93}}$ and $v_{\mathrm{Lu00}}$, hence their deviation from $v_x$, but not $v_{\mathrm{Mi79}}$, which is consistent with $v_x$.

The strong agreement between $v_x$ and $v_{\mathrm{Mi79}}$ strengthens our conclusion from Sect. 3.1.1 that the approximations made in deriving Eq. (20) are valid for representing the first-order form of $v$. It also means that under certain low-salinity conditions, Eq. (18) can be used directly with $\Delta_r H^{\ominus}$ values from the literature and no further fitting to quantify $v$ accurately with respect to calculations with an equilibrium model (PyCO2SYS). The strong agreement between $v_h^{\mathrm{Ta93}}$ and $v_{\mathrm{Lu00}}$ suggests that the extra equilibria and ionic strength effects can be adequately accounted for by adjusting the value of $b_h$ and without changing the form of the $t$-$\ln(f\mathrm{CO_2})$ relationship.

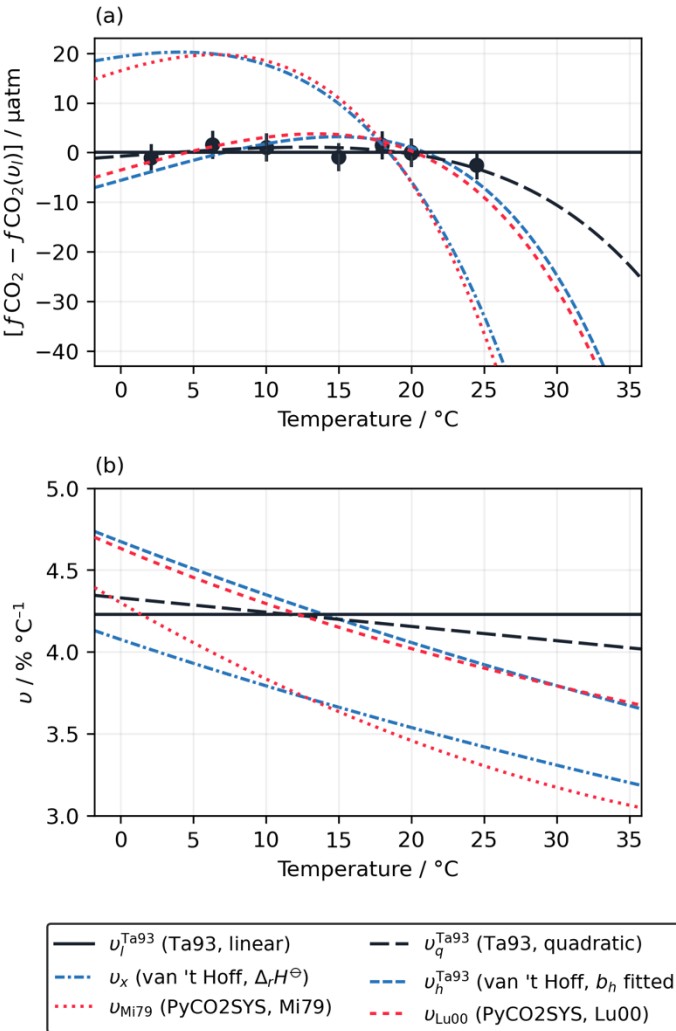

**Figure 2: (a) Variation of $f\text{CO}_2$ with temperature, normalised to the linear fit of Takahashi et al. (1993) ($v_l = 4.23\%\ °\text{C}^{-1}$), according to the measurements of Takahashi et al. (1993) (Ta93; filled circles with vertical $1\sigma$ error bars) and the different fitted and theoretical values of $v$ computed under the conditions of the Takahashi et al. (1993) experiment (Sect. 2.1.2). $v_x$ is based on Eq. (18), using the theoretical $b_h$ of 25288 J mol$^{-1}$ (Appendix A), while $v_h^{\text{Ta93}}$ uses Eq. (19) and the best-fit $b_h$ for the Takahashi et al. (1993) dataset of 28995 J mol$^{-1}$ (Sect. 2.1.3). (b) The values of $v$ calculated using each of the approaches shown in panel (a) as a function of temperature.**

It is unfortunate that the dataset of Takahashi et al. (1993) has a maximum temperature of 24.5 °C, because extra data points above 25 °C could unambiguously distinguish between the different forms of $v$. At lower temperatures, the differences between the forms are similar in size to the uncertainty of the measurements, so many more highly accurate measurements would need to be conducted to robustly distinguish between the forms (Fig. 2a). The Kanwisher (1960) measurements covered an even narrower temperature range from 11 to 21.5 °C, which is also insufficient to convincingly show non-linearity.

One published dataset does cover a wider temperature range, with seven measurements of a single sample from 5.05 to 35 °C (Lee and Millero, 1995). These measurements can be fitted well with both the linear and quadratic forms (Eqs. 5 and 6), with RMSDs of 1.8 μatm (linear) and 1.5 μatm (quadratic). Lee and Millero (1995) reported a constant $v_l$ of 4.26% °C$^{-1}$, stating that the difference between this and the value of Takahashi et al. (1993) was due to a different $T_C/A_T$ for the sample measured (0.85 versus ~0.92). As with the Takahashi et al. (1993) dataset, $v_h^{\text{LM95}}$ fits less well than either of $v_l$ or $v_q$, with an RMSD of 2.8 μatm, although again the fitted curve falls within the uncertainty window of the data points (±3 μatm). However, in this case, $v_h^{\text{LM95}}$ is completely different from the calculation with the Lueker et al. (2000) carbonic acid dissociation constants (Fig. 3). It also predicts significantly less deviation from $v_l$ or $v_q$ than in the Takahashi et al. (1993) case.

The fitted $b_h$ value of 30794 J mol$^{-1}$ for $v_h^{\text{LM95}}$ is unusual, falling outside the range fitted across the entire monthly mean OceanSODA-ETZH data product (25139 to 30731 J mol$^{-1}$; Sect. 2.4; Supp. Fig. 2a). Based on the $A_T$, $T_C$ and salinity

reported by Lee and Millero (1995), and with the carbonic acid dissociation constants of Lueker et al. (2000), we would have expected a $b_h$ of 29825 J mol$^{-1}$. Unlike Takahashi et al. (1993), which is not inconsistent with Eq. (20) but lacks data at high enough temperatures to convincingly support or reject it, the Lee and Millero (1995) dataset is clearly inconsistent with Eq. (20).

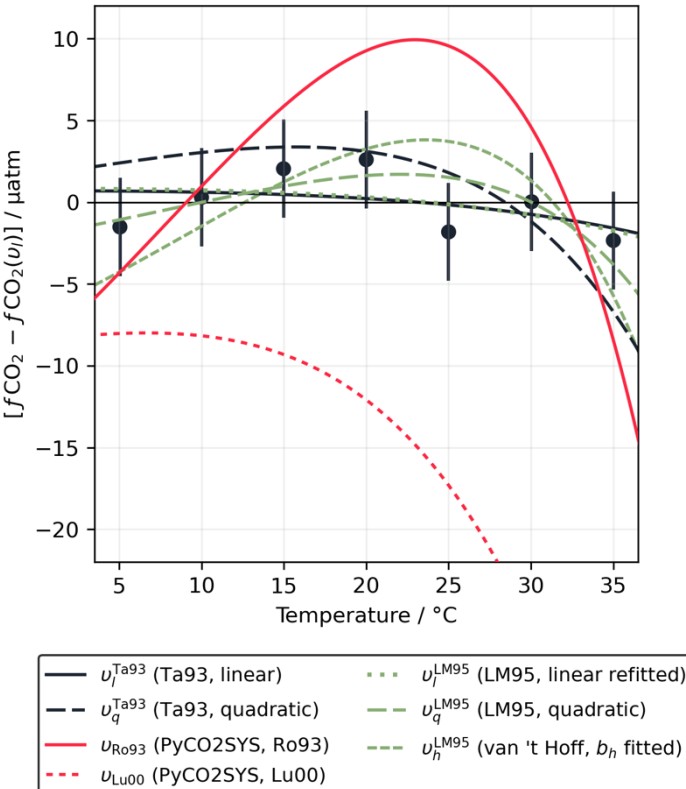

**Figure 3: Variation of $f$CO$_2$ with temperature, normalised to the linear fit of Lee and Millero (1995) ($v_l$ = 4.26% °C$^{-1}$), according to the measurements of Lee and Millero (1995) (LM95; filled circles with vertical 1$\sigma$ error bars) and the different fitted and theoretical values of $v$ computed under the conditions of the Lee and Millero (1995) experiment. $v_h^{LM95}$ uses Eq. (19) and the best-fit $b_h$ for the Lee and Millero (1995) dataset of 30794 J mol$^{-1}$ (Sect. 2.1.3). The line for $v_{Lu00}$ continues to decrease after exiting the figure, reaching –40 µatm at 35 °C.**

However, the Lee and Millero (1995) measurements are also inconsistent in a number of other ways not related to Eq. (20). We refitted the Lee and Millero (1995) dataset with the linear form (Eq. 5) and found a lower RMSD (1.6 µatm) by using Takahashi et al.'s (1993) $v_l$ value of 4.23% °C$^{-1}$. It appears that Lee and Millero (1995) minimised residuals in ln($f$CO$_2$) in their fit rather than minimising residuals in $f$CO$_2$; the latter approach returns almost exactly the Takahashi et al. (1993) value of 4.23% °C$^{-1}$. Minimising residuals in ln($f$CO$_2$) implicitly assumes that uncertainty in $f$CO$_2$ is a constant fraction of the $f$CO$_2$ value rather than an absolute fixed amount, but Lee and Millero (1995) reported the 'reproducibility' as ±3 µatm, not as a percentage. While at first glance it may seem an improvement that fitting the data consistently with the uncertainty brings the $v_l$ value in line with Takahashi et al. (1993), in fact this contradicts the reasoning of Lee and Millero (1995) themselves that a different value is expected because of the rather different $T_C/A_T$. For $T_C/A_T$ to be the same as that of Takahashi et al. (1993), either $T_C$ or $A_T$ would need to be adjusted by over 160 µmol kg$^{-1}$, which is far more than permitted by the measurement accuracy.

The Lee and Millero (1995) dataset is also not consistent with PyCO2SYS calculations. As mentioned above, the carbonic acid dissociation constants of Lueker et al. (2000) do not fit well at all, with an RMSD of 20 µatm (Fig. 3). While Lee and Millero (1995) claim that their data are 'thermodynamically consistent' with the constants of Roy et al. (1993) and Goyet and Poisson (1989), neither of these fits very well, both having an RMSD of ~6 µatm (Fig. 3). But they did acknowledge that more studies of the temperature sensitivity of the marine carbonate system are needed, pointing to possible problems with the Weiss (1974) formulation for CO$_2$ solubility, which is still used in PyCO2SYS (Humphreys et al., 2022).

In summary, while comparisons with the experimental datasets do not rule out Eq. (20), they are less convincing than the comparisons with PyCO2SYS calculations in Sect. 3.1.1. For the Takahashi et al. (1993) dataset, it is plausible that extra measurements at higher temperatures could have revealed significant deviations from linearity, as predicted independently by both PyCO2SYS and Eq. (20), although the linear $v_l$ still fits the data more closely than $v_h$ does. The Lee and Millero (1995) dataset, which does include higher temperature measurements, does not support this. This latter dataset does feature a number of other inconsistencies that could cast doubt on its accuracy, but alternatively, there may be some aspect of the equilibrium chemistry of $CO_2$ in seawater that is missing from or incorrectly parameterised within existing models such as PyCO2SYS. It is not possible to draw a reliable conclusion either way from only these two small datasets. We need a much bigger body of measurements to be able to robustly distinguish between the different possible forms of $v$.

### 3.1.3 Global data compilation

The Wanninkhof et al. (2022) compilation provides much more data (~2000 data points) with which to test the different temperature adjustment approaches, including across variations in salinity and carbonate chemistry ($A_T$ and $T_C$). The compilation consists of $fCO_2$ measured in discrete seawater samples at 20 °C (irrespective of the in situ temperature), along with $fCO_2$ measured at (or close to) the in situ temperature with a continuously flowing underway system at closely corresponding times and locations. Independent measurements of $A_T$ and $T_C$ are also available for many of the data points.

This dataset goes some way towards fixing the data scarcity problem mentioned in the previous Sect. 3.1.2, but it is still not a complete substitute for additional experiments like that of Takahashi et al. (1993). In particular, each sample has measurements of $fCO_2$ at only two different temperatures, so the form of the $t$-$fCO_2$ relationship between those two points cannot be determined for any individual sample. Instead, we must infer the suitability of different forms of $v$ from how well they work across the entire dataset.

We adjusted the $fCO_2$ at 20 °C from the discrete samples to the temperature of the in situ measurement using five different approaches: the linear ($v_l$) and quadratic ($v_q$) equations of Takahashi et al. (1993) (Sect. 2.1.2); the van 't Hoff form with $b_h$ constant as fitted to the Takahashi et al. (1993) dataset ($v_h^{Ta93}$; Sect. 2.1.3) and parameterised as a function of temperature, salinity and $fCO_2$ ($v_p$; Sect. 2.4); and calculated with PyCO2SYS using $T_C$ as the second known parameter and the carbonic acid dissociation constants of Lueker et al. (2000) ($v_{Lu00}$). To compare the accuracy of these adjustments, we computed the rolling mean of the differences between $fCO_2$ adjusted from 20 °C to the in situ temperature and $fCO_2$ measured directly at the in situ temperature for each approach, using a ±4 °C window (Fig. 4). Like Wanninkhof et al. (2022), we excluded outliers falling more than ±2 standard deviations from zero, which was 1.4% of the total data points.

For negative $\Delta t$ there are relatively few data (about 14% of the dataset), such that the different rolling mean lines overlap and mostly fall within the large uncertainty window (Fig. 4). It is not possible to meaningfully distinguish between the different approaches. All appear to overestimate $fCO_2(t_1)$ for $\Delta t$ from about –14 to –5 °C, by a similar amount to each other, with some falling just outside the uncertainty window of the rolling mean. The consistency between the different approaches here is expected, because they are most similar to each other in the corresponding $t_1$ range from about 10 to 20 °C (Fig. 2a). The minor deviations for negative $\Delta t$ are therefore probably due to the sparsity of data in this temperature range, which means that errors due to measurement uncertainty and imperfect co-location of samples are not averaged out. We would expect these deviations from zero to diminish were more data added to the compilation.

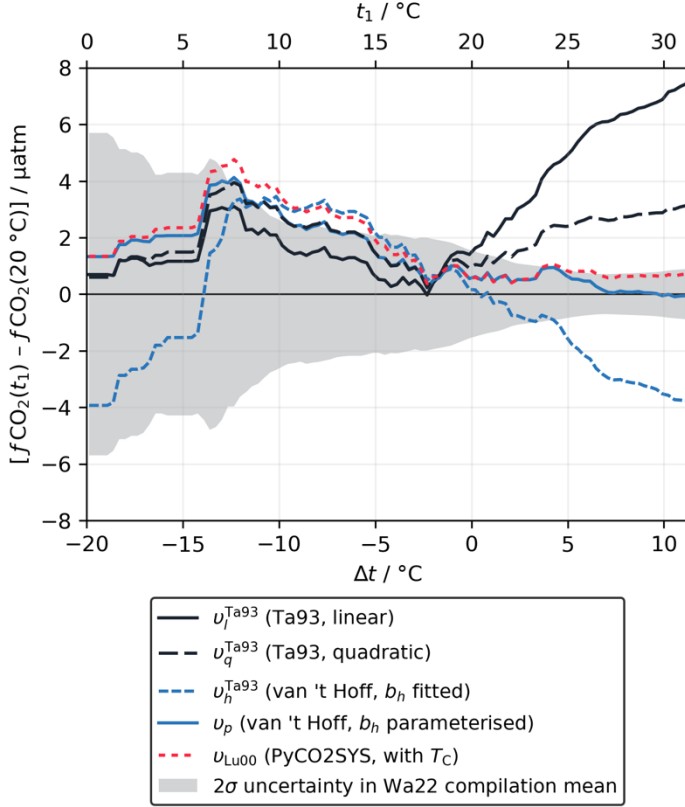

**Figure 4: Difference between $f\mathrm{CO}_2$ adjusted from 20 °C to the in situ temperature and a direct measurement at (or close to) the in situ temperature, shown as a rolling mean (window = ±4 °C) for each adjustment approach, for the data in the Wanninkhof et al. (2022) compilation (Wa22). The shaded area shows the $2\sigma$ uncertainty in the rolling mean, centred about zero, which is a function of the number of data points within each interval and virtually identical for every approach.**

The $\Delta t$ range above 0 °C contains the majority of the dataset (86%) and sufficient data for the approaches to diverge clearly from each other and for some to move well outside the uncertainty window (Fig. 4). Here, the linear form ($\upsilon_l$) significantly overestimates $f\mathrm{CO}_2(t_1)$ as $\Delta t$ increases. Wanninkhof et al. (2022) tried fitting $\upsilon_l$ directly to this data compilation to improve the accuracy of the adjustment but were not able to fix this issue. Instead, they concluded that calculations with a fully resolved carbonate system were still required for $\Delta t$ greater than about 1 °C, recommending the Lueker et al. (2000) carbonic acid dissociation constants with the Uppström (1974) borate:chlorinity for the best agreement ($\upsilon_{\mathrm{Lu00}}$). Like $\upsilon_l$, the quadratic form ($\upsilon_q$) overestimates $f\mathrm{CO}_2(t_1)$ for positive $\Delta t$. The error is about half of that for $\upsilon_l$, but still well outside the uncertainty window. The van 't Hoff form with constant $b_h$ ($\upsilon_h^{\mathrm{Ta93}}$) has a similar error to $\upsilon_q$, but with the opposite sign (i.e., $f\mathrm{CO}_2(t_1)$ is underestimated). However, the van 't Hoff form with a variable, parameterised $b_h$ ($\upsilon_p$) gives almost perfect agreement even for the largest $\Delta t$ – performing even better than the calculation with a fully resolved carbonate system ($\upsilon_{\mathrm{Lu00}}$), which also falls just within the uncertainty window.

This analysis shows that the van 't Hoff form is at least as capable as the other approaches of adjusting $f\mathrm{CO}_2$ to different temperatures across this data compilation. When variability in $b_h$ is accounted for by the parameterisation proposed in Sect. 2.4, which is completely independent of this data compilation, this approach ($\upsilon_p$) can be used for accurate $\Delta t$ adjustments of up to at least 10 °C. More data for negative $\Delta t$ are still essential to improve our confidence in this conclusion across the full temperature range found in the ocean.

Part of the motivation for the new approach to $\upsilon$ developed here (Sect. 2.1.3) was to avoid needing to know a second marine carbonate system parameter when adjusting $f\mathrm{CO}_2$ to different temperatures. $\upsilon_p$ with fixed $b_h$ does not meet this requirement, because like for $\upsilon_l$ and $\upsilon_q$, the adjusted $f\mathrm{CO}_2$ values are not accurate enough. Initially it appears that $\upsilon_p$ does meet the requirement – its only inputs are $f\mathrm{CO}_2$, temperature and salinity (Eq. 35), and it is equally or more accurate than

calculations with PyCO2SYS (Fig. 4). But arguably, the need for a second parameter has been sidestepped rather than truly eliminated. The parameterisation for $b_h$ works because, to first order, not only $b_h$ (and thus $\upsilon_h$) but also $fCO_2$ are closely correlated with the ratio $T_C/A_T$, and less sensitive to the absolute values of either $T_C$ or $A_T$ (Goyet et al., 1993; Wanninkhof et al., 2022). In other words, the $fCO_2$ term in the parameterisation acts as a proxy for $T_C/A_T$, implicitly providing the information about $T_C$ and $A_T$ necessary to calculate $\upsilon$.

It would be possible to parameterise $\upsilon_l$ and $\upsilon_q$ across the ocean like we have done for $\upsilon_h$ (i.e., $\upsilon_p$), which would improve their agreement with the data compilation too. However, we have already established that there is no theoretical reason to expect the $t$-ln($fCO_2$) relationship to be linear, especially at higher $t$, and Wanninkhof et al. (2022) tried fitting $\upsilon_l$ to this dataset directly without satisfactory results. Further benefits of $\upsilon_h$ over $\upsilon_q$ are discussed in Sect. 3.2 on uncertainty propagation.

## 3.2 Experimental uncertainty

We performed Monte-Carlo simulations of the Takahashi et al. (1993) experiment (Sect. 2.2.2) to determine the uncertainty in $\upsilon$ computed from that dataset using the linear ($\upsilon_l$; Eq. 5), quadratic ($\upsilon_q$; Eq. 6) and van 't Hoff ($\upsilon_h^{\mathrm{Ta93}}$; Eq. 19) fits.

Were the linear fit a valid representation of the $t$-ln($fCO_2$) relationship, then the $1\sigma$ uncertainty in experimentally determined $\upsilon_l$ would be 0.035% °C$^{-1}$. The biggest contributor (0.028% °C$^{-1}$ on its own) is the precision of the $fCO_2$ measurement, with 0.020% °C$^{-1}$ from systematic bias in $fCO_2$ alone. Systematic biases can affect $\upsilon$ because it is the gradient of the natural logarithm of $fCO_2$ (Eq. 1), which is a non-linear transformation. Temperature uncertainties have a much smaller impact, with 0.001% °C$^{-1}$ uncertainty in $\upsilon$ arising from $t$ precision alone and no effect from $t$ bias. Propagating through to exp($Y_l$) for a $\Delta t$ of +1 °C gives an uncertainty of just under 0.04% (Fig. 5; Supp. Fig. 4). Following Eq. (4), the percentage uncertainty in exp($Y$) propagates through to the same percentage uncertainty in the adjusted $fCO_2$ value at $t_1$, so this calculation is most relevant for understanding how uncertainties in $\upsilon$ affect temperature-adjusted $fCO_2$ values.

With the quadratic fit (Eq. 6), the $1\sigma$ uncertainty in the slope ($b_q$) is 0.127% °C$^{-1}$, in the squared term ($a_q$) $41.2 \times 10^{-6}$ °C$^{-2}$, and there is a covariance between these of $\sigma(a_q, b_q) = -51 \times 10^{-9}$ °C$^{-3}$. Propagating through to $\upsilon_q$ and exp($Y_q$) gives an uncertainty that is similar to that for the linear fit in the middle of the $t$ range but is up to four times greater in colder and warmer waters (Fig. 5; Supp. Fig. 4). It is essential to include the covariance term for propagation; failure to do so wrongly inflates the final uncertainty.

The van 't Hoff fit has a $1\sigma$ uncertainty in $b_h$ of 216.4 J mol$^{-1}$. Propagating through to $\upsilon_h^{\mathrm{Ta93}}$ and exp($Y_h^{\mathrm{Ta93}}$) gives a similar uncertainty to that for $\upsilon_l$, decreasing slightly with increasing temperature (Fig. 5; Supp. Fig. 4).

The uncertainties calculated here, for $\Delta t = +1$ °C, are all lower than the "climate quality" target uncertainty for seawater $fCO_2$ measurements of 0.5% proposed by GOA-ON (Newton et al., 2015). This means that this uncertainty component alone would not cause an $fCO_2(t_1)$ value with $\Delta t$ up to ±1 °C to breach this target, regardless of which form was correct, although it is probably not small enough to ignore in a complete uncertainty budget. Also, its contribution scales roughly in proportion with $\Delta t$, so for greater $\Delta t$ this component would become a greater part of the overall uncertainty in $fCO_2(t_1)$.

Why do the different approaches have different uncertainties and how do we choose which approach and corresponding uncertainty to use? The linear and van 't Hoff forms both appear to have similarly low uncertainties, but the temperature adjustments that they predict differ from each other by more than the apparent uncertainty windows, especially above 25 °C (Fig. 2). The quadratic form has a completely different uncertainty profile with apparently much greater uncertainty at the edges of the temperature range (Fig. 5). The assumption implicit within any uncertainty calculation that the model being fitted to the data is valid, and that all its adjustable parameters are truly unknown and necessary, is key to resolving this.

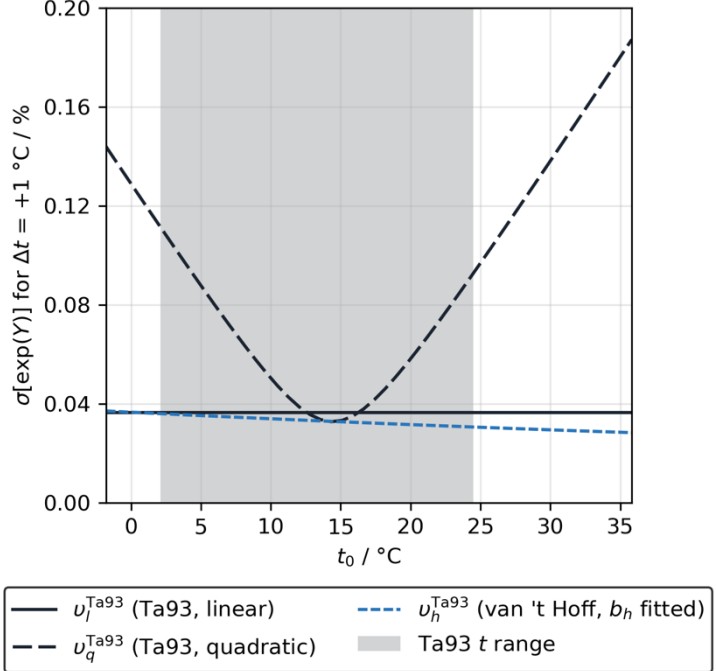

**Figure 5: 1σ uncertainty in exp($Y$) for Δ$t$ = +1 °C, which is equal to the percentage uncertainty in $f$CO$_2$ at $t_1$ caused by uncertainty in $v$, due to experimental uncertainties in $t$ and $f$CO$_2$ in the Takahashi et al. (1993) dataset only and fitted with the linear (dotted line; Eq. 5) and quadratic (dashed line; Eq. 6) forms as well as the fitted van 't Hoff form (solid line; Eq. 19). The shaded area shows the range of $t$ from the Takahashi et al. (1993) experiment, while the full $t$ axis range matches OceanSODA-ETZH. The uncertainty in $v$ itself as a function of temperature has a virtually identical pattern to that shown here for exp($Y$) (e.g., an uncertainty in exp($Y$) of 0.04%, with Δ$t$ = +1 °C, corresponds to an uncertainty in $v$ of about 0.04% °C$^{-1}$; Supp. Fig. 4).**

As we have shown in Sect. 3.1, it is not consistent with existing models of the marine carbonate system for $v$ to be constant and independent from temperature, so the uncertainty calculated for $v_l$ is valid only under this premise. In other words, the assumptions that "$v$ is constant" and "$v$ varies with temperature" are mutually incompatible. This is how $v_l$ and $v_h^{Ta93}$ can deviate from each other by more than their apparent uncertainties.

It makes sense that $v_q$ has greater uncertainty than $v_l$ and $v_h^{Ta93}$ because $v_q$ has an extra coefficient to be constrained by the same number of data points. Given more data at higher $t$, $v_q$ could plausibly be fitted to a shape almost identical to $v_h$, yet still it would appear to have greater uncertainty for this reason. When using $v_q$, we accept that there is some curvature in the $t$-ln($f$CO$_2$) relationship, but we implicitly assume that we do not know what form that curvature takes, other than it can be represented by some second-order polynomial. The adjustable parameters that represent the curvature are thus constrained empirically only by the available data, leading to much higher apparent uncertainty towards its edges. But we now know that, to first order, $v$ should follow a particular curvature that can be represented with only one adjustable parameter ($b_h$); it is not an unknown that must be found empirically. The lower uncertainty in $v_h^{Ta93}$ and exp($Y_h^{Ta93}$) relative to $v_q$ and exp($Y_q$) is therefore more realistic (Fig. 5), with the caveat that more experimental work is still needed to determine whether the expected relationship from Eq. (20) can be produced in laboratory measurements (Sect. 3.1.2).

While small, this uncertainty component is probably not negligible for the most accurate analyses. More independent reproductions of the Takahashi et al. (1993) experiment, ideally including temperatures up to 35 °C, could reduce this component into a negligible term. However, this component reflects only the experimental uncertainties in $f$CO$_2$ and temperature. To use this value as the total uncertainty in $v$ and/or exp($Y$) would be to assume that the single sample measured by Takahashi et al. (1993) is representative of the global ocean. It does not account for spatiotemporal variability in $v$, which we discuss in Sect. 3.3.

### 3.3 Spatial and temporal variability

#### 3.3.1 Variability in $v_{\text{Lu00}}$

The time-averaged theoretical value of surface ocean $v_{\text{Lu00}}$ in OceanSODA-ETZH (Gregor and Gruber, 2021) follows a latitudinal distribution with lower values near the equator (Fig. 6a). The range in $v_{\text{Lu00}}$ is more than 15% of its mean, from less than 4.0% °C$^{-1}$ to over 4.6% °C$^{-1}$. The pattern is dominantly driven by temperature. The other variables ($A_{\text{T}}$, $T_{\text{C}}$ and salinity) together act in the opposite sense, slightly reducing the latitudinal gradient in $v_{\text{Lu00}}$. This can be seen from the distribution of

$v_h^{\text{Ta93}}$, which uses the constant $b_h$ found by fitting to the Takahashi et al. (1993) dataset (Sect. 2.1.3), and which has a similar but stronger latitudinal gradient (Fig. 6b). A counterintuitive consequence of this is that, were the Takahashi et al. (1993) experiment to be repeated systematically across the globe, fitting with a linear equation at each location, then this would return the opposite pattern of $v_l$ to what we see in Fig. 6a: higher values near the equator and lower near the poles. The variability in $v_l$ as determined following Takahashi et al. (1993) is independent from the original in situ temperature of the sample, so the

experiment would reveal variability in $v$ only due to the non-thermal parameters.

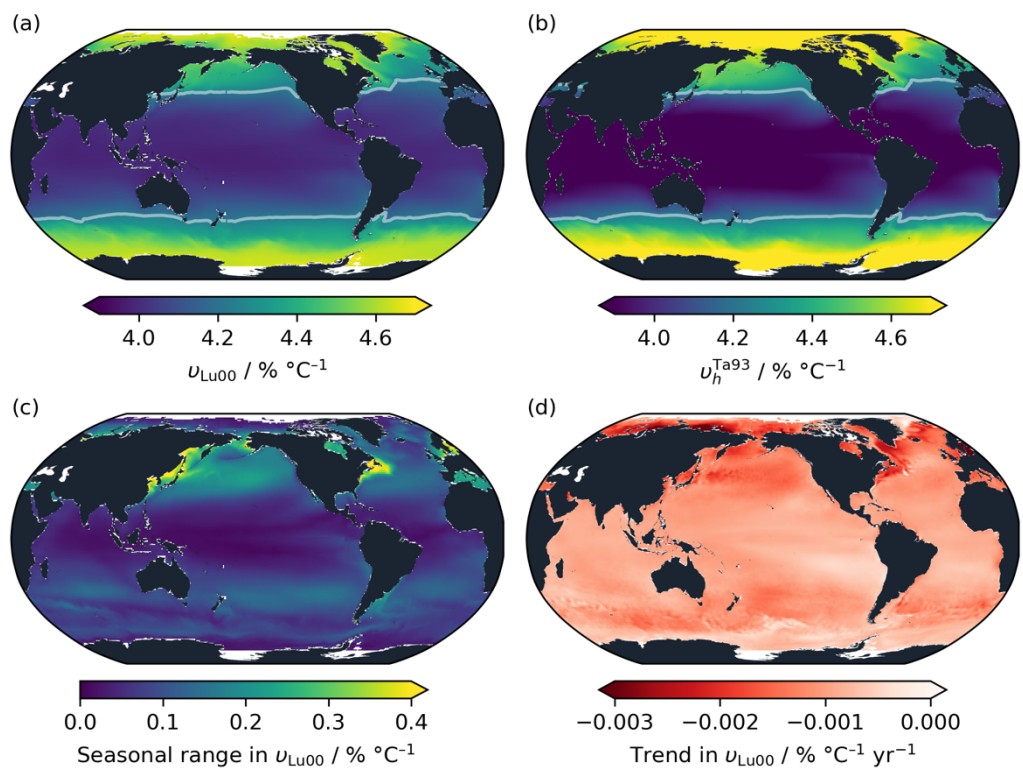

**Figure 6: Spatial and temporal variability in $v$ in OceanSODA-ETZH using the Lueker et al. (2000) parameterisation of the carbonic acid dissociation constants. (a) Theoretical $v_{\text{Lu00}}$ calculated from $A_{\text{T}}$, $T_{\text{C}}$ and auxiliary variables, averaged through time across the entire OceanSODA-ETZH dataset. (b) $v_h^{\text{Ta93}}$ calculated from OceanSODA-ETZH temperature using the $b_h$ value fitted to the**
**Takahashi et al. (1993) measurements, again averaged through time. (a) and (b) are on the same colour scale and the white contour in both shows the Takahashi et al. (1993) linear fit value of $v_l$ = 4.23% °C$^{-1}$. (c) Mean seasonal range and (d) multi-decadal trend in $v_{\text{Lu00}}$.**

The seasonal range in $v$ is a similar size to, although slightly less than, the spatial range in its time-averaged value. It is less than 0.2% °C$^{-1}$ across most of the open ocean, reaching a maximum of just over 0.4% °C$^{-1}$, or around 10% of the mean, in
some near-coastal areas with strong seasonal variability in temperature and/or $T_{\text{C}}/A_{\text{T}}$ (Fig. 6c).

Together, the spatial and seasonal variability in $v_{\text{Lu00}}$ has a standard deviation of 0.23% °C$^{-1}$. This could be considered to represent the $1\sigma$ uncertainty in using a constant $v_l$ globally, keeping in mind that it has strong regional systematic biases rather than being randomly distributed. It is significantly greater than the apparent experimental uncertainty in $v_l$ or $v_h^{\text{Ta93}}$ of around 0.04% °C$^{-1}$ (Sect. 3.2). Experiments like that of Takahashi et al. (1993) should be sufficiently accurate to detect the

variability in $v$ if conducted at different locations around the global ocean. But it also means that unaccounted-for spatiotemporal variability would lead to several times greater uncertainty in $v$ and $\exp(Y)$ in practice than was computed in Sect. 3.2.

On multi-decadal timescales, $v$ has been decreasing globally at an average rate of 0.001% $°C^{-1}$ $yr^{-1}$ (Fig. 6d). The decrease has been primarily driven by increasing $T_C/A_T$ with a smaller contribution from warming of the surface ocean. This

trend is consistent with that identified by Wanninkhof et al. (2022), and it should lead to a minor decrease in the temperature dependency of surface ocean $fCO_2$ while atmospheric $CO_2$ levels continue to rise.

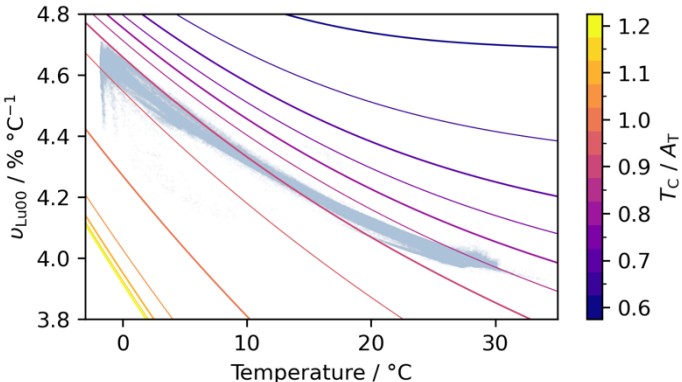

**Figure 7: Each coloured line shows how $v_{Lu00}$ varies with temperature at a constant $T_C/A_T$, with $T_C$ always at the median value of the monthly mean OceanSODA-ETZH data product (i.e., 2045 µmol $kg^{-1}$). $T_C/A_T$ increases from 0.6 (dark purple, top right) to 1.2**
**(yellow, bottom left). The light blue points in the background show all monthly mean data from OceanSODA-ETZH.**

Wanninkhof et al. (2022) pointed out that $v$ is greater at lower temperatures and lower at lower $T_C/A_T$, and that in the global surface ocean, temperature and $T_C/A_T$ are to first order negatively correlated (largely due to the influence of temperature on $CO_2$ solubility), leading them to have to opposing effects on $v$ and thus reducing its variability. We find that this effect does reduce spatial variability in $v$, but at most by 30% relative to a constant $T_C/A_T$ scenario. For a temperature change from –1.7 to

30 °C, the main body of the OceanSODA-ETZH monthly mean data product has a range in $v_{Lu00}$ from about 3.97% to 4.60% $°C^{-1}$, whereas at a constant $T_C/A_T$ of 0.9, $v_{Lu00}$ varies from 3.85% to 4.72% $°C^{-1}$ (Fig. 7). However, this partial cancellation of the effect of temperature on $v$ by $T_C/A_T$ is not relevant for making temperature adjustments; it does not mean that the temperature sensitivity of $v$ can be ignored or is any less important. This is because adjustments are made on single samples under constant $T_C/A_T$. The sensitivity of $v$ to temperature for any individual sample is not lessened by the negative

correlation between temperature and $T_C/A_T$ observed across the global dataset. In other words, coupled temperature-$T_C/A_T$ variations in the surface ocean reduce the spatiotemporal variability in $v$ under in situ conditions, but do not affect by how much $v$ changes with temperature for a particular sample. Consequently, the apparent reduced variability in $v$ is relevant only for small (e.g., $\Delta t < 1$ °C) temperature adjustments.

### 3.3.2 Components of $v$

Variability in $v$ through space and time arises primarily from the temperature-dependence of the equilibrium constants of the marine carbonate system. Across the contemporary global surface ocean, the biggest contributor to $v$ is $K_0'$, the solubility constant for $CO_2$, which typically represents 58–89% of total $v$ (Fig. 8). The carbonic acid equilibrium constants $K_1^*$ and $K_2^*$ contribute a similar size effect on $v$ as each other and slightly less than $K_0'$. However, the $K_1^*$ contribution is always negative and the $K_2^*$ positive, so they partly cancel each other out. Smaller contributions are made by the borate (9–21%) and water (1–

8%) equilibria ($K_B^*$ and $K_w^*$). All other equilibrium constants have a negligible influence on $v$ in the contemporary ocean, although they might become important at pH values closer to their p$K$.

Each parameterisation of the carbonic acid dissociation constants gives a different $v$ distribution but there are some similarities (Fig. 8). We could not identify a link between the functional forms of the parameterisations (i.e., which

combinations of $t$ and salinity terms were used) and the distributions shown in Fig. 8; rather, the type of seawater, specific dataset(s) and methodology used to determine the constants were the main controls.

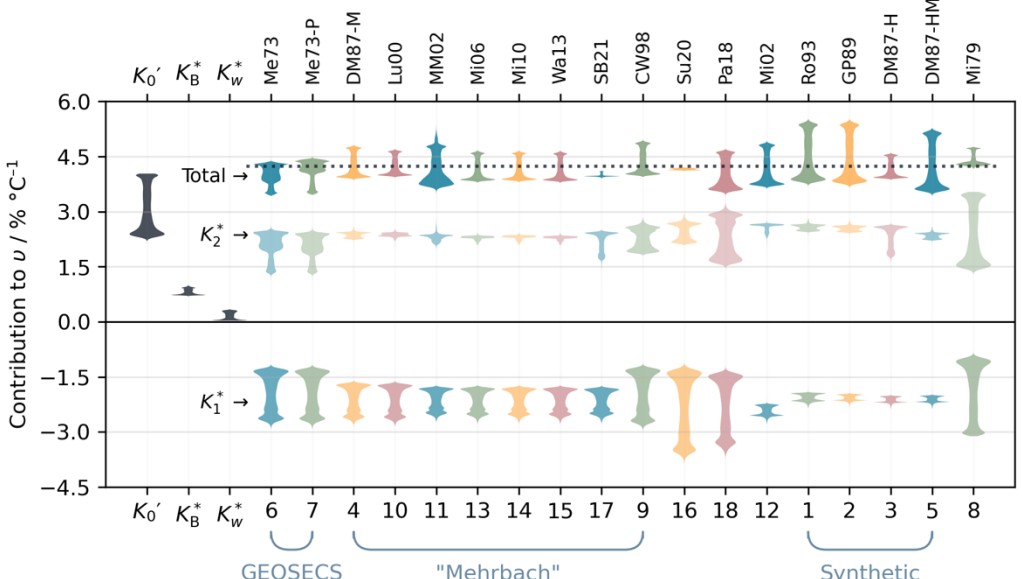

**Figure 8: Violin plot of the contributions of each equilibrium system to $v$ as distributed across the monthly averaged OceanSODA-ETZH data product. From the left, the first three violins show the contributions of the $CO_2$ solubility constant ($K_0'$) and the borate ($K_B^*$) and water ($K_w^*$) equilibria. The remaining columns show the contributions of the carbonic acid dissociation constants $K_1^*$ and $K_2^*$ as well as the total $v$ for each parameterisation in PyCO2SYS v1.8.3. The horizontal dashed line through the "Total" group is at the Takahashi et al. (1993) linear fit value ($v_l = 4.23\%$ °C$^{-1}$). The colours are to help the eye track the groups vertically; they do not have any further meaning. The lower horizontal axis shows the option code within PyCO2SYS for the carbonic acid constants, while the upper axis shows an abbreviation of the corresponding study: 1/Ro93 = Roy et al. (1993); 2/GP89 = Goyet and Poisson (1989); 3/DM87-H, 4/DM87-M and 5/DM87-HM = refits of various combinations of Hansson (1973a, b) and Mehrbach et al. (1973) by Dickson and Millero (1987); 6/Me73 and 7/Me73-P = GEOSECS fits of Mehrbach et al. (1973); 8/Mi79 = Millero (1979), for freshwater; 9/CW98 = Cai and Wang (1998); 10/Lu00 = Lueker et al. (2000); 11/MM02 = Mojica Prieto and Millero (2002); 12/Mi02 = Millero et al. (2002); 13/Mi06 = Millero et al. (2006); 14/Mi10 = Millero (2010); 15/Wa13 = Waters and Millero (2013); 16/Su20 = Sulpis et al. (2020); 17/SB21 = Schockman and Byrne (2021); 18/Pa18 = Papadimitriou et al. (2018).**

In Fig. 8, the two options labelled "GEOSECS" (6 and 7; Me73 and Me73-P) both use the measurements of Mehrbach et al. (1973), as parameterised by Peng et al. (1987) (because Mehrbach et al. did not provide parameterised functions for their measurements); curiously, total $v$ with these options has a roughly opposite spatial distribution to all of the other options, with the lowest $v$ found at high latitudes. Although the GEOSECS options also use different $K_B^*$ and $K_w^*$ parameterisations from all the other options (Humphreys et al., 2022), which modulates the pattern, the inversion is caused by $K_1^*$ and $K_2^*$. In any case, these options are present in (Py)CO2SYS to reproduce calculations from the GEOSECS era and are not intended for use with modern measurements. Aside from these pair, all other parameterisations based at least in part on the Mehrbach et al. (1973) dataset (labelled "Mehrbach") have similar distributions for $K_1^*$, $K_2^*$ and total $v$. The SB21 parameterisation (Schockman and Byrne, 2021) consists of new, spectrophotometric measurements of the product $K_1^*K_2^*$ which were paired with the Waters and Millero (2013) $K_1^*$ to find $K_2^*$, which resulted in overall virtually zero variability in total $v$. This low variability in total $v$ is echoed by the Su20 parameterisation (Sulpis et al., 2020), which is based on field observations where $A_T$, $T_C$, pH and $f CO_2$ were measured simultaneously, but the low variability is arrived at in a different way, with rather different distributions for the individual $K_1^*$ and $K_2^*$ effects. However, this does not mean that these parameterisations support the use of a constant, temperature-insensitive $v$, but rather just that the effects of temperature and seawater chemistry (e.g., $A_T/T_C$) on in situ $v$ cancel each other out across the surface ocean more completely than they do for Lueker et al. (2000) (Fig. 7). Indeed, neither of SB21 and Su20 can be fitted to the Takahashi et al. (1993) dataset very well (Supp. Fig. 2). Parameterisations Pa18 (Papadimitriou et al., 2018), focused on high-salinity, low-temperature sea ice brines, and Mi02 (Millero et al., 2002), based on field measurements, both have similar variability in $v$ as the "Mehrbach" group.

There is a noticeable difference in the distributions, especially for $K_1^*$, when moving to the group of parameterisations based on synthetic seawater (Dickson and Millero, 1987; Goyet and Poisson, 1989; Ro93, GP89, DM87-H and DM87-HM; Roy et al., 1993). This suggests that some components of natural seawater that were not replicated in the synthetic seawater used in their experiments may affect the apparent temperature sensitivity of the equilibrium constants being measured, and thus $\upsilon$. For example, the synthetic seawater recipes used by Goyet and Poisson (1989) and Roy et al. (1993) both did not include any borate, unlike real seawater (Lee et al., 2010; Uppström, 1974). Suspected interactions between borate and carbonate species (Mojica Prieto and Millero, 2002) were not explicitly modelled when evaluating the equilibrium constants, so their effects would be implicitly included within the $K_1^*$ and $K_2^*$ values in real seawater, but not in the synthetic. Dissolved organic matter could also have an equivalent effect. Ultimately, any dissolved species that reacts in an equilibrium with water (i.e., any weak acid or base) or with any component of $T_C$ (i.e., $CO_2(aq)$, $HCO_3^-$ or $CO_3^{2-}$) could affect $\upsilon$.

The parameterisation of Mi79 is also shown for completeness, although its distribution is not meaningful because it is intended for zero-salinity freshwater (Millero, 1979).

The parameterisations of $K_1^*$ and $K_2^*$ differ from each other sufficiently that, given the high accuracy with which we can measure $f\mathrm{CO}_2$ and thus experimentally determine $\upsilon$ (Sect. 3.2), more experiments like that of Takahashi et al. (1993) but also including a wider range of temperatures (i.e., at least up to 35 °C) would be a powerful tool to help distinguish between the different parameterisations of the equilibrium constants (Supp. Fig. 2) and other options (e.g., borate:chlorinity; Humphreys et al., 2022) used when solving the marine carbonate system. Taking this a step further, experimental data like that of Takahashi et al. (1993), Woosley (2021) and Woosley and Moon (2023) could be incorporated when generating these parameterisations. These data constrain the derivatives of the equilibrium constants with respect to temperature, which the parameterisations are not normally optimised for. Constraining these derivatives in the fit would mean the parameterisation can more accurately calculate derivative properties like $\upsilon$. Such an approach is common in other fields, such as the Pitzer model for chemical activities in aqueous solutions (Pitzer, 1991). There, different measurable properties of the solution constrain the derivatives of fitted coefficients with respect to temperature, pressure and composition, so a single set of coefficients can be found that can accurately calculate all these properties (e.g., Archer, 1992).

At present, including derivative properties in parameterisations would be limited by the scarcity of measurements like those of Takahashi et al. (1993). One possible alternative would be to use data like that compiled by Wanninkhof et al. (2022) and datasets where the marine carbonate system has been overdetermined (i.e., more than two parameters measured) in the field to better parameterise the equilibrium constants (e.g., Sulpis et al., 2020). A practical challenge with this would be that $f\mathrm{CO}_2$ measurements are archived in different compilations (e.g., SOCAT; Bakker et al., 2016) than the other carbonate system variables (e.g., GLODAP; Lauvset et al., 2024) and it is not straightforward to match them back together. However, even were this overcome, this approach would be less useful for parameterising equilibrium constants than using dedicated experiments where well-characterised samples are manipulated under a range of different conditions. In the ocean, temperature is correlated with most other properties, so it would be difficult to isolate which of several correlated drivers was responsible for observed variability. Using these field data to generate parameterisations would also mean that they could no longer be used for independent validation. A better approach would be to reserve field data compilations for testing parameterisations that are based on more carefully controlled laboratory experiments. Combining new experimental measurements with field data in this way and interpreting the results through the lens of the new theoretical basis for $\upsilon$ developed in this study will lead to more accurate temperature adjustments of seawater $f\mathrm{CO}_2$ measurements and contribute towards a more internally consistent understanding of the marine carbonate system.

### 3.3.3 Propagation through to temperature adjustments

Expressions for $v$ such as $v_l$, $v_q$ and $v_h$ are of interest because they can be used to adjust $fCO_2$ data to different temperatures without knowing a second carbonate system parameter. The variability in $v$ discussed so far in Sect. 3.3 is therefore most important in terms of how it affects $\exp(Y)$ and consequent uncertainty and biases in the adjusted $fCO_2$ values if ignored.

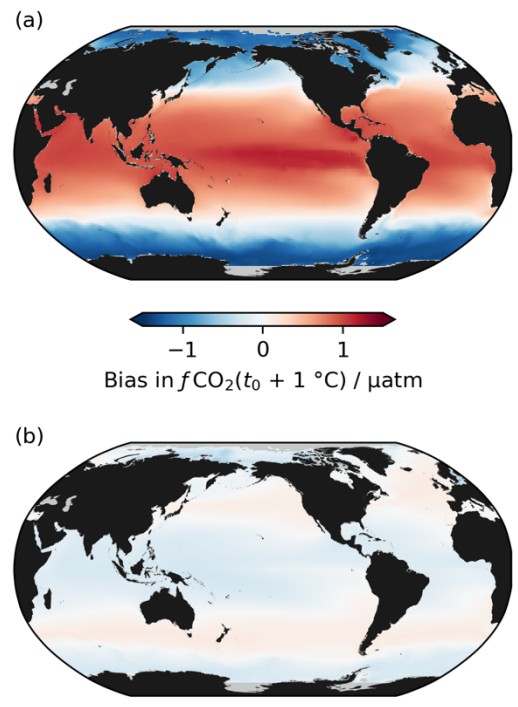

(a)

Bias in $fCO_2(t_0 + 1\ °C)$ / µatm

(b)

**Figure 9: Bias in adjusted $fCO_2$ for a 1 °C adjustment ($\Delta t = +1$ °C) relative to the change in $fCO_2$ calculated from $A_T$ and $T_C$ with PyCO2SYS and the Lueker et al. (2000) parameterisation of the carbonic acid dissociation constants when the adjustment is applied using (a) the Takahashi et al. (1993) constant $v_l$, or (b) the van 't Hoff form with parameterised $b_h$ (Sect. 2.4). The colour scale is the same for (a) and (b).**

After a 1 °C warming adjustment from mean surface ocean conditions in OceanSODA-ETZH, both the $v_l$ and $v_p$ forms have a global mean bias (relative to the change calculated from $A_T$ and $T_C$) of less than 0.1 µatm, which is negligible. That the global mean bias is close to zero for $v_l$ is a coincidence, due to Takahashi et al. (1993) using seawater with roughly average $v_{\mathrm{Lu00}}$ (Fig. 6a). But there are stronger regional biases in $v_l$, with 95% of the residuals falling in the range from –1.3 to +1.1 µatm, leading to a global RMSD of 0.8 µatm, or equivalently 0.24% (Fig. 9a). The van 't Hoff form with parameterised $b_h$ ($v_p$) has

much lower regional biases, with a global RMSD of 0.16 µatm (0.04%) and with 95% of the residuals falling from –0.26 to +0.24 µatm (Fig. 9b). The Takahashi et al. (1993) quadratic fit ($v_q$) and the van 't Hoff form with constant $b_h$ fitted to the Takahashi et al. (1993) dataset ($v_h^{\mathrm{Ta93}}$) (both not shown) had negligible global mean biases and intermediate regional biases, with RMSDs of around 0.4 µatm.

The experimental $1\sigma$ uncertainty in $\exp(Y_l)$ and $\exp(Y_h^{\mathrm{Ta93}})$ fitted to the Takahashi et al. (1993) dataset was around

835 0.04% for a 1 °C warming adjustment in both cases (Sect. 3.2). For $v_l$, the additional uncertainty from unaccounted-for spatiotemporal variability, plus the fact that the $t$-$\ln(fCO_2)$ relationship is not linear, is therefore four times greater than the experimental uncertainty. For $v_h$, the experimental and spatiotemporal uncertainties are about equal in size. Combining these components gives a $1\sigma$ uncertainty in adjusted $fCO_2$ of around 0.06% for $v_p$, or 0.24% for $v_l$. A 1 °C adjustment with either approach could therefore still meet the GOA-ON climate quality goal of 0.5% (Newton et al., 2015), if other uncertainty

components (e.g., in the original $fCO_2$ measurement at $t_0$) were small enough. However, the spatiotemporal component of the uncertainty in $v_l$ results from a known bias with strong regional patterns. Such biases should be eliminated by using a more realistic form of $v$ (e.g., $v_h$ rather than $v_l$ or $v_q$, and $v_p$ or similar to account for variability in $v_h$) rather than incorporating them

into an overall uncertainty value. But if applying $v_l$ globally then 0.24% is a reasonable value for the component of uncertainty due to a +1 °C adjustment. It is important to remember that this component is a systematic bias with strong regional patterns when combining it with other components that may be independent for each measurement, because these two types of uncertainty have very different statistical consequences when propagated through to other calculations.

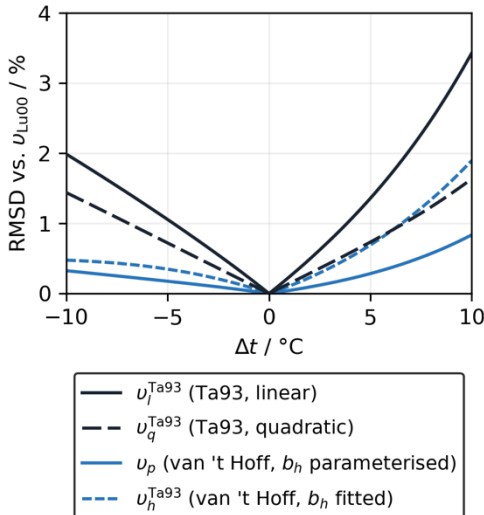

**Figure 10: Effect of $\Delta t$ on the global RMSD of the residuals in the OceanSODA-ETZH data product between $f\mathrm{CO_2}$ adjusted by $\Delta t$ using the value calculated from $A_\mathrm{T}$ and $T_\mathrm{C}$ ($v_\mathrm{Lu00}$) and each of $v_l$, $v_q$, $v_h$ (all fitted to Ta93) and $v_p$.**

The global biases and the amplitude of spatiotemporal variability scale approximately, but not exactly, with $\Delta t$ (Fig. 10). The $v_p$ approach has the lowest RMSD throughout the $\Delta t$ range, while $v_l$ has the greatest. $v_q$ and $v_h$ have intermediate RMSDs, with that for $v_h$ significantly lower than $v_q$ for negative $\Delta t$. The RMSDs are generally greater for positive $\Delta t$ than for negative $\Delta t$, which emphasises the need to better understand the temperature sensitivity of $f\mathrm{CO_2}$ and the wider marine carbonate system, particularly at higher temperatures (e.g., Sect. 3.1.1; Woosley, 2021). For a $\Delta t$ of 10 °C, as might be used in studies unravelling the drivers of seasonal $f\mathrm{CO_2}$ changes, the adjusted $f\mathrm{CO_2}$ would have an uncertainty due to spatiotemporal variability of around 0.8% for $v_p$, outside the GOA-ON climate quality goal (0.5%) but well within the weather quality goal of 2.5% (Newton et al., 2015). However, with $v_l$, this component of the uncertainty in adjusted $f\mathrm{CO_2}$ would be over 3%, also exceeding the weather quality goal.

**4 Conclusions**

Seawater $f\mathrm{CO_2}$ data can be adjusted to different temperatures using either Eq. (2) or Eq. (4) (which are mathematically identical) with an appropriate expression for $Y$, which is $v$ integrated over the temperature range of interest. The new approach proposed here uses $Y_h$ as defined in Eq. (21), which has one unknown coefficient ($b_h$). The value of $b_h$ can be found by fitting to experimental data where available (e.g., Table 1), but spatiotemporal variability in $b_h$ should be accounted for, for example with the parameterisation in Eq. (35). This approach, as well as the earlier linear and quadratic approaches of Takahashi et al. (1993), have been built into the PyCO2SYS software as of v1.8.3.

There is no theoretical reason to expect a linear relationship between temperature and $\ln(f\mathrm{CO_2})$ under isochemical conditions (constant $A_\mathrm{T}$ and $T_\mathrm{C}$), as was proposed empirically by Takahashi et al. (1993). Rather, calculations of $v$ with PyCO2SYS and a theoretical analysis based on the equations governing the marine carbonate system and the van 't Hoff equation each independently suggest that $\ln(f\mathrm{CO_2})$ should be proportional to $1/t_\mathrm{K}$ to first order (Sect. 3.1.1). Although this means that the $t$-$\ln(f\mathrm{CO_2})$ relationship does seem to be approximately linear within the temperature range of 2.1 to 24.5 °C measured by Takahashi et al. (1993), it should diverge rapidly from linearity especially at temperatures above 25 °C, although

there are some discrepancies with the very few existing laboratory data which require more measurements to resolve (Sect. 3.1.2). Nevertheless, the new form of the relationship between temperature and $f\mathrm{CO_2}$ thus proposed here ($v_h$ with a parameterised $b_h$ coefficient, i.e., $v_p$; Eq. 19 and Sect. 2.4) agrees better with field-based datasets than any other approach previously used to adjust seawater $f\mathrm{CO_2}$ to different temperatures (Sect. 3.1.3).

The parameterisation for $b_h$ (and thus $v_p$) is based on a specific set of carbonate system parameterisations. We know that these are not perfect (e.g., Woosley and Moon, 2023; Carter et al., 2024), so the $b_h$ parameterisation is an intermediate solution. It does agree very well with the available field data (Wanninkhof et al., 2022) with which it can be tested (and from which it is independent), but it may need to be updated when new data and/or more accurate parameterisations of the marine carbonate system become available. For some applications, accuracy may be improved by developing parameterisations of $b_h$ that are optimised for any particular region(s) of interest.

The uncertainty in $v$ depends on the method being used to compute it. The Takahashi et al. (1993) experiment gave a $1\sigma$ uncertainty of about 0.04% °C$^{-1}$ for the linear form (Sect. 3.2), but this does not account for variability in $v$ with temperature and seawater composition, nor for the fact that the relationship between temperature and $\ln(f\mathrm{CO_2})$ is not linear. A $1\sigma$ in $v$ of 0.23% °C$^{-1}$ is therefore more appropriate if using this linear approach (Sect. 3.3.1), which corresponds to 0.24% for $f\mathrm{CO_2}$ adjusted by +1 °C (Sect. 3.3.3). This uncertainty can be reduced by using the new form of $t$-$\ln(f\mathrm{CO_2})$ relationship developed here and parameterising its fitted constant ($b_h$) as a function of temperature, salinity and $f\mathrm{CO_2}$ (Sect. 2.4), which gives an uncertainty of 0.06% °C$^{-1}$ for $f\mathrm{CO_2}$ after the same +1 °C adjustment (Sect. 3.3.3). While these uncertainties are probably not large enough to have major consequences for global air-sea $\mathrm{CO_2}$ flux budgets, they are not negligible and may be important in regional studies. They should be incorporated into uncertainty budgets and their systematic components should be eliminated by using a more meaningful and accurate form to model the $t$-$\ln(f\mathrm{CO_2})$ relationship, like that proposed here. However, the uncertainty budget is still incomplete: we cannot propagate uncertainties from the equilibrium constants of the marine carbonate system through to $v$, because those uncertainties are not yet adequately defined (Orr et al., 2018). But if $v$ can be fitted directly to measurements like those of Takahashi et al. (1993), then the uncertainties in the equilibrium constants can be ignored when adjusting $f\mathrm{CO_2}$ values to different temperatures.

More measurements like those conducted by Takahashi et al. (1993) are still needed to empirically confirm whether the $v_h$ form, and indeed calculations from $A_T$ and $T_C$, are sufficiently accurate, especially above 25 °C where the deviation from $v_l$ is expected to be greatest. If further measurements appear to confirm the high-temperature linearity observed by Lee and Millero (1995) then there must be either a major error in a parameterisation or missing component of the equilibrium model for seawater chemistry implemented in software tools like PyCO2SYS at higher temperatures or some technical problem with the experimental approach that means it does not measure quite what we think. More measurements could show whether discrepancies between $v_p$ and $v_{Lu00}$ are due to needing a more sophisticated equation for $v_h$ with extra fitted parameters, or whether they represent inaccuracies in the Lueker et al. (2000) and other carbonate system parameterisations. In principle, we can measure the effect of temperature on $f\mathrm{CO_2}$ accurately enough to distinguish between these different parameterisations. Coupling such new experiments with other measurements such as pH on the same samples may also help to understand and rectify wider discrepancies in equilibrium models of seawater chemistry. Together with field measurements for independent validation, and using a theoretical basis to interpret the results rather than working only with empirical fits, such a holistic approach should not only help to better understand $v$ and its application to adjusting $f\mathrm{CO_2}$ data to different temperatures, but also lead to a more accurate and internally consistent model of the marine carbonate system.

**Appendix A: Standard enthalpies of reaction**

The reactions corresponding to the equilibrium constants of interest are (Humphreys et al., 2022)

$$\mathrm{CO_2(g)} \overset{K_0'}{\rightleftharpoons} \mathrm{CO_2(aq)} \tag{A1}$$

$$CO_2(aq) + H_2O \overset{K_1^*}{\rightleftharpoons} HCO_3^- + H^+ \tag{A2}$$

$$HCO_3^- \overset{K_2^*}{\rightleftharpoons} CO_3^{2-} + H^+ \tag{A3}$$

The standard enthalpies of formation ($\Delta_f H^{\ominus}$) for the relevant species are given in Table A1.

Using Hess's law, the standard enthalpies of reaction for Eqs. (A1)-(A3), as used in Eq. (18), are therefore

$$\Delta_r H_0^{\ominus} = \Delta_f H^{\ominus}[CO_2(aq)] - \Delta_f H^{\ominus}[CO_2(g)] = -19\,720 \text{ J mol}^{-1} \tag{A4}$$

$$\Delta_r H_1^{\ominus} = \Delta_f H^{\ominus}(HCO_3^-) + \Delta_f H^{\ominus}(H^+) - \Delta_f H^{\ominus}[CO_2(aq)] - \Delta_f H^{\ominus}(H_2O) = 9\,134 \text{ J mol}^{-1} \tag{A5}$$

$$\Delta_r H_2^{\ominus} = \Delta_f H^{\ominus}(CO_3^{2-}) + \Delta_f H^{\ominus}(H^+) - \Delta_f H^{\ominus}(HCO_3^-) = 14\,702 \text{ J mol}^{-1} \tag{A6}$$

Summing Eqs. (A4)-(A6) as required by Eq. (18) and propagating the uncertainties listed in Table A1 gives a value of $\Delta_r H_2^{\ominus} - \Delta_r H_0^{\ominus} - \Delta_r H_1^{\ominus} = 25288 \pm 98$ kJ mol$^{-1}$.

**Table A1: Standard enthalpies of formation at 298.15 K following Ruscic and Bross (2023).**

| Species | $\Delta_f H^{\ominus}$ / J mol$^{-1}$ |
|---|---|
| $CO_2(g)$ | $-393\,476 \pm 15$ |
| $CO_2(aq)$ | $-413\,196 \pm 19$ |
| $H_2O(l)$ | $-285\,800 \pm 22$ |
| $HCO_3^-(aq)$ | $-689\,862 \pm 39$ |
| $CO_3^{2-}(aq)$ | $-675\,160 \pm 53$ |
| $H^+(aq)$ | 0 (by definition) |

**Code availability**

The code for the updated version of PyCO2SYS (v1.8.3) is freely available on GitHub and archived on Zenodo (Humphreys
et al., 2024) and can be installed in Python using either pip or conda (via conda-forge). The code used for the rest of the analysis in this manuscript is available via Zenodo (https://doi.org/10.5281/zenodo.13492773).

**Acknowledgements**

This manuscript was inspired by discussions during the "Series of workshops on surface ocean $p$CO$_2$ observations, synthesis and data products" held at the Flanders Marine Institute (VLIZ) in November 2023. We are indebted to the authors of the
930 previous studies that collected and made available the data used here, especially the measurements of Takahashi et al. (1993), the compilation of Wanninkhof et al. (2022) and the OceanSODA-ETZH data product (Gregor and Gruber, 2021). We thank Rik Wanninkhof and Fiz Pérez for providing reviews and Mario Hoppema for editorial comments which helped to improve the manuscript.

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
