# Peer review of "Temperature effect on seawater $fCO_2$ revisited: theoretical basis, uncertainty analysis, and implications for parameterising carbonic acid equilibrium constants"

_EGUsphere, 2024_

## Referee Comment (RC1)

Title: Temperature effect on seawater ƒCO2 revisited: theoretical basis, uncertainty analysis, and implications for parameterising carbonic
Author(s): Matthew P. Humphreys
MS No.: egusphere-2024-626
MS type: Research article
Iteration: Initial submission

General comments:
The paper studies the sensitivity of seawater carbon dioxide fugacity (ƒCO2) to temperature because it is crucial for the accurate measurements needed in constructing global carbon budgets and understanding the variability of CO2 flux between air and sea. So far, the normalization or correction for temperature impact on fCO2 has been done using an experimental determination (Takahashi et al. 1993) and alternatively using the equations of the carbonate system in seawater (i.e. Wanninkof et al. 2020). The author discusses that the two are not fully compatible or that it is possible to improve their small discrepancies. The authors present a new approach based on marine carbonate system equations and the van 't Hoff equation that shows a different proportional relationship between ln(ƒCO2) and temperature. It is argued that this approach is consistent with experimental and field data, and offers lower uncertainty in the temperature sensitivity of ƒCO2. Their results may have more important implications for regional budgets and significant temperature adjustments, they are unlikely to affect global air-sea CO2 flux budgets.

In my opinion the manuscript presents some very important errors in the theoretical approach and the improvement it proposes to evaluate the effect of temperature on fCO2 does not improve the proposal of Wanninkof et al. (2022). One might think that it could be presented as a useful alternative anyway, but my doubt is whether this could lead many readers to some confusion.

Minor coments
Line 47.- "it is driven by a global mean $\Delta$ƒCO2 of less than 10 $\mu$atm" This need a citation.

Line 64. Also, is use in the analysis of GOBMs to decomposed the biological and the temperature effects over the $pCO_2$ seasonal cycle (i.e. Rodgers et al. 2023)

Linea 71-99.- The author strongly emphasizes that the derivative of $fCO_2$ with respect to temperature cannot be theoretically linear. This is somewhat obvious, but it does not exclude that given a temperature interval, such as the one described in the APPENDIX of the Takahashi et al. (1993) article ranging from 2 to 24, the inverse function of Kelvin temperature and centigrade temperature are correlated with an R2=0.9995, which implies an error of less than 2% in the estimation of the dependent variable by not using an inverse function of kelvin temperature. The additional exercise performed by these same authors of fitting to a quadratic function decreases this error to 0.04%. Therefore, I believe that the use of the linear function is overly criticized, although it is true that the use of the Takahahshi et al.' factor should be restricted to the 2 to 24°C range and for relatively small temperature changes so as not to result in somewhat biased $fCO_2$ estimates.

Line 98-99.- The author is aware that it is possible to use an approximate alkalinity to obtain a new $fCO_2$ at another temperature without producing an error greater than 2 µatm. Since the salinity is known, any climatology can generate an alkalinity with an error of ±20 µmol/kg , it would generate

a new $fCO_2$ with an error of less than 0.3 µatm for a temperature change of 10°C. Therefore, the use of CO2SYS as proposed by Wanninkoff et al 2022 is more than sufficient.

Line 105. Please, show or evaluate the "big differences".

Line 108.- "thus indicating some deficiency with the measurements of Takahashi et al. 1993". The weaknesses of the Takahashi et al. (1993) measurements are not evaluated throughout the article. After all, they are the ones used throughout the article as a reference for other types of parameterizations.

Line 111: "we aim to provide this missing theoretical basis by developing a new functional form for how $fCO_2$ and thus $u$ vary with temperature' Are you saying that the measurements and statistical adjustments made by Takahashi et al. 1993 show a lack of theoretical basis? Honestly, they simply made a linear fit because for that temperature range it would be practically identical to a fit vs $1/t_k$.

Line 162-163. "Takahashi et al. (1993) did not give a theoretical basis for either of the forms (linear and quadratic; Eqs. 5 and 6) that they fitted to their dataset nor did they give any reason to choose one over the other." It does not seem very necessary to provide any kind of theoretical basis when the high-quality measures of Takahashi et al. 1993, shows that pCO2 values correlate with r2=0.9999. The same $r^2$ that would come out using the CO2SYS with the same configuration shown in the article.

Line 177 "In typical seawater, the approximations in Eqs. (12) and (13) are more than 99% accurate for $T_C$ and more than 97% accurate" I honestly believe that this is where the author makes a big mistake because this approach has important consequences. The $C_T/A_T$ ratio is key in his approximation. It is true that for $C_T/A_T$ ratios<0.9 the approximation is quite correct and the van't Hoff model would work well, but for $C_T/A_T$ values>0.9 equation 11 is going to present very low carbonate concentration (even lower than $CO_2$ concentrations and with high pCO_2 values generating very high biases when van't Hoff model is applied). The strange thing is that this is not indicated until very late in the article and is somewhat overlooked.

Line 300-303. "At each grid point, we computed the mean across all years separately for each month for temperature, salinity, $A_T$ and $T_C$. We then used PyCO2SYS to calculate $fCO_2$ from these variables at 50 evenly spaced temperatures from −1.8 to 35.83 °C (i.e., the range of the OceanSODA-ETZH data product) at each month and grid point. Next, we fit Eq. (19) to the generated $t$ and $fCO_2$ data to find the best fitting $b_h$ value at each point." To be the key part of the way to obtain the equation 35 I think it is poorly explained. Especially because the equation first obtains 50 fCO2 data by applying the CO2SYS equations, and then adjusting it to the van`t Hoff equation which would only be applicable with high precision with low $C_T/A_T$ values, i.e. with low $fCO_2$ values which implies a low error in its adjustment. On the contrary, for high $C_T/A_T$ values (high $fCO_2$ values) the application of equation 19 already deviates from equation 11, and the biases generated amplify the errors in the $fCO_2$ estimates using $b_h$. (Fig 3b).

Line 305-66 "The coefficients and their variance-covariance matrix are provided in the Supplementary Information (Supp. Tables 1-2)." It would be more informative for the reader to include the uncertainties of each of the coefficients and their level of significance.

Line 352. Legend Figure 1 "**a) Variation of $fCO_2$ with temperature according to the measurements of Takahashi et al. (1993)**". The legend is misleading. The variations of $fCO_2$ with temperature are not described, but the anomaly of $fCO_2$ with respect to that estimated using various parameterizations, including the two proposed by Takahashi et al. 1993.

Line 413. Figure 2. How can it be explained that if the coefficient obtained from van't Hoff called '$b_h$ fitted' shows a behavior so different, and worse, from the $b_h$ parameterized van't Hoff coefficient considering that it is derived from that one. And on the other hand, the proposed parameterized $b_h$ enhancement is no better than the application of $u_{Lu00}$ as proposed by Wanninkof et al 2022.

Line 433 Figure 3. Figure 3a clearly shows that the proposed estimate ($b_h$ van`t Hoff) of $fCO_2$ changes due to temperature changes does not improve on the more accurate alternative of direct application of CO2SYS when the $C_T/A_T$ ratio>0.95. The legend to Figure 3b shows the RMSE of $b_h$, but the units is $K^{-1}$ and not the erroneously written $\mu atm^{-1}$.

Line 435 "the $b_h$ fit is less than 1 $\mu$ atm". The $b_h$ unit is $J\ mol^{-1}$ such as is indicated in Line 202.

Line 449 "parameterisations. However, while these issues might contribute a component of the discrepancy – i.e., the main pattern with RMSD ~1 $\mu$ atm seen for $T_C/A_T$ less than ~0.95 – there is no reason to expect their influence to be correlated with $T_C/A_T$, so they cannot be the entire explanation." How is it not possible that the author himself seems unaware of the limitations of equation 19 which proceeds from a strong simplification of equation 11? Just at $C_T/A_T$ values below ~0.95, the term removed from equation 1 becomes determinant because the denominator tends to zero. We are in the environment of the first equivalence point where carbonate concentrations are equal to $CO_2$ concentrations.

Line 452-457 "Inaccuracies in the approximations $A_x$ and/or $T_x$, used in generating Eq. (19), likely also play a role at higher $T_C/A_T$. The fraction of $T_C$ comprised of $[CO_2(aq)]$, which is ignored in $T_x$, increases with $T_C/A_T$, while the fraction of non-carbonate alkalinity, ignored in $A_x$, decreases with increasing $T_C/A_T$. Consequently, the approximation that $[HCO_{3-}]^2/[CO_3^{2-}]$ is constant across different temperatures (Eq. 16), which emerges from the definitions of $A_x$ and $T_x$ (Eqs. 12-15), becomes less accurate with increasing $T_C/A_T$. The $T_x$ approximation may be the problem here rather than $A_x$, because the RMSD of the $b_h$ fit Is positively correlated with the error in $T_x$ but negatively correlated with the error in $A_x$ (Supp. Fig. 3)." Clearly the problem is the denominator of equation 11, and certainly it is the 'Inaccuracies in the approximations Ax and/or Tx, used in generating Eq. (19),' There is not the slightest doubt and this calls into question the usefulness of $b_h$ and in the background of this whole article. Why use an alternative parameterization to avoid using CO2SYS which is always going to be less accurate even if we have indeterminacies in the equilibrium constants?

Line 520 "But we now know that $u$ should follow a particular curvature that can be represented with only one adjustable parameter ($b_h$)" Sure? The equation 19 is a simplification of equation 11. On the one hand, the equation 11 contains more factors than the apparent equilibrium constants of the marine carbonate equilibrium and therefore does not have to follow exactly the van't Hoff equation. On the other hand, the apparent (or empirical) carbonic acid constants ($K_1$ and $K_2$) as well as the $CO_2$ saturation ($K^o$) contain more summands than the one given by the van't Hoff equation which are polynomial functions of the kelvin temperature.

Line 575 (Figure 5) and594-597. "The SB21 parameterisation (Schockman and Byrne, 2021) consists of new, spectrophotometric measurements of the product $K_{1*}K_{2*}$ which …$k_{2*}$, which resulted in overall virtually zero variability in total $u$. This low variability in total $u$ is echoed by the Su20 parameterisation (Sulpis et al., 2020), which is based on field observations where $A_T$, $T_C$, pH and $fCO_2$ were measured simultaneously, but the low variability is arrived at in a different way, with rather different distributions for the individual $K_{1*}$ and $K_{2*}$ effects". Interesting figure. Obviously, the probability curves represent the spatial distribution of the ocean surface and this is strongly dependent on latitude. But looking at a relative perspective when comparing one set of constants with others, it is interesting to note that both the Schockman &Byrne 'Mehrbach' option (SB21) and Sulpis 2020 (Su20) show virtually no spatial variability with values very close to that estimated by Takahashi et al. (1993). This is an interesting aspect of this study.

Line 664 Conclusions. This epigraph is too long, and in some parts, it is rather a new discussion.

References

Rodgers, K. B., Schwinger, J., Fassbender, A. J., Landschützer, P., Yamaguchi, R., Frenzel, H., et al. (2023). Seasonal variability of the surface ocean carbon cycle: a synthesis. *Global Biogeochemical Cycles*,37, e2023GB007798. https://doi.org/10.1029/2023GB007798

---

## Referee Comment (RC2)

Review with figure:
EGUsphere
Review of: Temperature effect on seawater ƒCO2 revisited: theoretical basis, uncertainty analysis, and implications for parameterising carbonic acid equilibrium constants
Matthew P. Humphreys
Reviewer Rik Wanninkhof, AOML

Matthew Humphreys provides a physical chemical basis to the frequently used empirical relationship of fCO2 with temperature along with an thorough uncertainty analysis. He has incorporated the results into the PyCO2SYS software. The temperature relationship is frequently used to correct surface water fCO2 measurements to *in situ* temperatures. Based on the uncertainty analysis presented, the uncertainty for this correction can be reduced from 0.24 % to 0.06 %.

The manuscript is exhaustive and laid out in a coherent fashion. It is well-referenced and thoroughly researched. Non-the-less it is a challenging read and final results and conclusions are not always clearly laid out. For instance, the equations to determine the temperature dependence are not that clearly laid out. That is, it would be useful if an additional equation was added after Eqns. 19-21 where the numerical values constants for $b_h$ and R were included.

A central premise of the work is that the proper functionality for the temperature dependence is 1/T, following the van t'Hoff relationship, rather than t which is well-founded but the fit of a 1/T relationship to the experimental data of Takahashi et al. 1993 is worse than a linear fit with t. At the end of the manuscript the author correctly states that over a narrow range the functionality of 1/T versus t is very similar.

As pointed out by others, the temperature dependence will depend on the bicarbonate and carbonate concentrations of the seawater. This is also described in this manuscript (eqn12-18) but never emphasized. The important point is that no single simple equation can predict fCO2 solely based on temperature but that information on TA and DIC needs to be implicitly included. It might be worth emphasizing that knowledge or estimate of TA and DIC will decrease the uncertainty in the ln(fCO2-T) relationship.

One aspect that is not emphasized is why this empirical single fCO2-temperature dependence developed by Takahashi based on a single seawater composition "works" for the surface ocean as described in Wanninkhof et al. 2022:" The theoretical temperature dependence expressed as ∂ln(fCO2w)/∂T = B0 + B1 T shows a stronger dependency at lower temperatures, and weaker dependency at low TA/DIC values (Fig. 4). These conditions often go hand-in-hand in the surface ocean. That is, surface waters of the world's oceans have lower temperatures and lower TA/DIC at higher latitudes such that the two factors will oppose each other."

For experimental data and verification the paper relies heavily on the pCO2-temperature relationship of Takahashi that was derived from 1 seawater sample and 8 measurements at 7 temperatures from 2-25 C. The paper also assumes that deviations from a linear trend are likely at higher temperatures and suggests, correctly, that more systematic studies have to be undertaken. While several groups have undertaken such studies, they have failed to publish it in

the open literature. A notable exception is that of Lee and Millero (1995) that provides measurements from 5-30 C and obtains a linear dependence very similar to the of Takahashi et (1993) (see attached figure) particularly if the sample at 5 Č is omitted that is questionable (Lee personal comm, 1996). Of note are that the 30 C value falls in line with other points, but that the temperature dependence using a polynomial fit is twice that of Takahashi.

Comments by line number

Line 11: omit "purely"

Line 25 and beyond : Besides the constants the concentrations of the species will impact the relationship as well

Line 47: global mean $\Delta fCO2 \approx$ -5 µatm (Fay et al., 2024)

Line 50 and beyond: "minimum accuracy of 0.5 %. " since $\Delta fCO2$ is often the quantity of interest it is commonly expressed as < 2 µatm for "climate quality" rather than a %

Line 55: state that the surface seawater is equilibrated with an enclosed headspace and the headspace is measured. That is fCO2 is fundamentally a gas phase property

Line 59: This criterium is for SOCAT dataset flags of A and B; SOCAT accepts all fCO2 measurements

Line 65: Of note is that fCO2 is also measured on discrete samples at fixed temperature usually 20 Č by select groups. In these case conversion to fCO2 at in situ temperatures is much greater.

Line 73: replace: "measured" by "reported"

Line 93: insert "υ, " after "calculating"

Line 109: Note that Takahashi et al. 2009 provide the equation as well including the integrated form

Sections 2.1 and 2.3: Very nice description of physical basis and uncertainties

Line 139: note that 8 measurements were taken at 7 temperatures.

Line 331: Note Lee and Millero (1996) have a measurement at 30 C that could be used to spot-check the difference between υl and υq above about 25 C

Line 335-340: the issue that the proposed fit does not do as well as the original is a significant point, even if both fall within the calculated uncertainty. Again the Lee and Millero (1996) measurements could shed some light on the fundamental issue if the proposed equations have shortcomings.

Line 427: Section 3.1.3 I found this section confusing in part because the previous sections discuss fitting with experimental data while this section discusses using the Lueker constants. I had expected that the author would use the experimental data presented in Table 3 of the paper of Lueker et al. which would be a good test of their parameterizations as the tests were done at several temperatures and varying fCO2, TA and DIC.

Line 455: This point could be empathized "Consequently, the approximation that [HCO3–]2/[CO32–] is constant across different temperatures (Eq. 16), which emerges from the definitions of Ax and Tx (Eqs. 12-15), becomes less accurate with increasing TC/AT.

Line 675 and before:" but this does not account for variability in υ through space and time" [while this is explained in the introduction], strictly speaking υ varies with temperature and chemical composition, not space and time.

Lee, K., and F. J. Millero, 1995: Thermodynamic studies of the carbonate system in seawater. *Deep-Sea Res.*, **42,** 2035-2061.

Takahashi, T., and Coauthors, 2009: Climatological mean and decadal change in surface ocean pCO₂, and net sea-air CO₂ flux over the global oceans. *Deep -Sea Res II*, **2009,** 554-577.

Note should be L&M, 1995

---

## Author Comment (AC1)

*General comments:*

*The paper studies the sensitivity of seawater carbon dioxide fugacity (fCO2) to temperature because it is crucial for the accurate measurements needed in constructing global carbon budgets and understanding the variability of CO2 flux between air and sea. So far, the normalization or correction for temperature impact on fCO2 has been done using an experimental determination (Takahashi et al. 1993) and alternatively using the equations of the carbonate system in seawater (i.e. Wanninkof et al. 2020). The author discusses that the two are not fully compatible or that it is possible to improve their small discrepancies. The authors present a new approach based on marine carbonate system equations and the van 't Hoff equation that shows a different*

*proportional relationship between ln(fCO2) and temperature. It is argued that this approach is consistent with experimental and field data, and offers lower uncertainty in the temperature sensitivity of fCO2. Their results may have more important implications for regional budgets and significant temperature adjustments, they are unlikely to affect global air-sea CO2 flux budgets.*

> ➢ My thanks to the reviewer for reading the manuscript and providing comments. I agree with the summary above.

*In my opinion the manuscript presents some very important errors in the theoretical approach and the improvement it proposes to evaluate the effect of temperature on fCO2 does not improve the proposal of Wanninkof et al. (2022). One might think that it could be presented as a useful alternative anyway, but my doubt is whether this could lead many readers to some confusion.*

> ➢ A number of the points presented as problems by the reviewer are either misunderstandings or do not take into account the evidence presented. This must stem from the original manuscript not having been written clearly enough, so I thank the reviewer for highlighting the points of potential confusion, and have revised the manuscript aiming to make these aspects clearer, in particular with the restructuring of the first part of the R&D (Sect. 3.1).

*Minor coments*

*Line 47.- "it is driven by a global mean ΔfCO2 of less than 10 µatm" This need a citation.*
> ➢ Value updated with citation.

*Line 64. Also, is use in the analysis of GOBMs to decomposed the biological and the temperature effects over the pCO2 seasonal cycle (i.e. Rodgers et al. 2023)*
> ➢ Thanks, citation added.

*Linea 71-99.- The author strongly emphasizes that the derivative of fCO2 with respect to temperature cannot be theoretically linear. This is somewhat obvious, but it does not exclude that given a temperature interval, such as the one described in the APPENDIX of the Takahashi et al. (1993) article ranging from 2 to 24, the inverse function of Kelvin temperature and centigrade temperature are correlated with an R2=0.9995, which implies an error of less than 2% in the estimation of the dependent variable by not using an inverse function of kelvin temperature. The additional exercise performed by these same authors of fitting to a quadratic function decreases this error to 0.04%. Therefore, I*

*believe that the use of the linear function is overly criticized, although it is true that the use of the Takahahshi et al.' factor should be restricted to the 2 to 24°C range and for relatively small temperature changes so as not to result in somewhat biased fCO₂ estimates.*

> ➤ I agree completely with the restrictions that the reviewer suggests should be applied to using the Takahashi et al. adjustment approach. However, these restrictions are not a solution: there is a lot of ocean outside the 2 to 24 °C range, and many applications where adjustments over large temperature ranges are required. Fixing this problem requires understanding how the $t$-$f$CO₂ relationship should behave, which is the main aim of this paper.

*Line 98-99.- The author is aware that it is possible to use an approximate alkalinity to obtain a new fCO2 at another temperature without producing an error greater than 2 µatm. Since the salinity is known, any climatology can generate an alkalinity with an error of ±20 µmol/kg , it would generate a new fCO2 with an error of less than 0.3 µatm for a temperature change of 10°C. Therefore, the use of CO2SYS as proposed by Wanninkoff et al 2022 is more than sufficient.*

> ➤ The method that the reviewer suggests (estimating alkalinity from salinity) is not proposed by Wanninkhof et al.; that study only considers cases where TA and DIC had been directly measured alongside $f$CO₂. Indeed, the complete method suggested by the reviewer has not to my knowledge been peer reviewed nor robustly assessed. A calculation with PyCO2SYS confirms that an uncertainty of 20 µmol/kg in alkalinity would indeed propagate through to an uncertainty of less than 0.3 µatm in $f$CO₂ adjusted by 10 °C (conditions: alkalinity = 2250 µmol/kg, $f$CO₂ = 400 µatm, temperature = 15 → 25 °C, salinity = 35). However, this uncertainty budget is far from complete. It ignores the uncertainties in the equilibrium constants of the CO₂ system which are used in these calculations. While these uncertainties are very poorly known, using the set proposed by Orr et al. (2018) inflates the total uncertainty in corrected $f$CO₂ by 100 times, to around 16 µatm. The adjusted $f$CO₂ value can also vary by ±10 µatm or more depending on which parameterisations are selected for the equilibrium constants. I have added extra text to the introduction here to point out this problem (although using a slightly different set of numbers; lines 100-113).
>
> ➤ More fundamentally, the method suggested by the reviewer does not deliver any scientific understanding of how temperature and $f$CO₂ are related, but rather takes more of an engineering approach with (Py)CO2SYS as a black-box tool, which doesn't provide any conceptual insight into how we expect the system to behave.

*Line 105. Please, show or evaluate the "big differences".*
> ➤ Added a value based on Fig. 3a from McGillis and Wanninkhof (2006).

*Line 108.- "thus indicating some deficiency with the measurements of Takahashi et al. 1993". The weaknesses of the Takahashi et al. (1993) measurements are not evaluated throughout the article. After all, they are the ones used throughout the article as a reference for other types of parameterizations.*

➢ The only 'deficiency' being referred to here is the lack of data above 25 °C. Other than the uncertainty in the alkalinity value, which is mentioned, I am not aware of any other weaknesses of these measurements that should be discussed here.

Line 111: *"we aim to provide this missing theoretical basis by developing a new functional form for how fCO2 and thus υ vary with temperature'* *Are you saying that the measurements and statistical adjustments made by Takahashi et al. 1993 show a lack of theoretical basis?*
➢ The choice to fit the measurements with linear and quadratic equations lacks a theoretical basis. The fitted forms are empirical, having been selected because they appeared to fit the data, without understanding why.

*Honestly, they simply made a linear fit because for that temperature range it would be practically identical to a fit vs 1/tk.*
➢ While the two fits are indeed very similar for the temperature range of the experiment, Takahashi et al. (1993) did not appear to consider a $1/t_k$ form, nor did they show that they knew this would be theoretically justified.

Line 162-163. *"Takahashi et al. (1993) did not give a theoretical basis for either of the forms (linear and quadratic; Eqs. 5 and 6) that they fitted to their dataset nor did they give any reason to choose one over the other."* *It does not seem very necessary to provide any kind of theoretical basis when the high-quality measures of Takahashi et al. 1993, shows that pCO$_2$ values correlate with r2=0.9999. The same r2 that would come out using the CO$_2$SYS with the same configuration shown in the article.*
➢ There are a few reasons why a theoretical basis is desirable, which I have attempted to make clearer in the revised manuscript: predictability, uncertainty propagation, and philosophical.
➢ Predictability: the linear fit does indeed perform very well within the range of the Takahashi et al. dataset, but it disagrees significantly with CO2SYS outside that range. However, the new form proposed here, still fitted only to the Takahashi dataset, agrees very well with CO2SYS even far outside of the range of the measurement data.
➢ Uncertainty propagation: as discussed in Section 3.2, accurate uncertainty propagation requires a meaningful and well-fitting model.
➢ Finally, the philosophical case is that, it is useful to know why something works, not just that it works (same as for the second bullet point of my response to 'lines 98-99' above).

Line 177 *"In typical seawater, the approximations in Eqs. (12) and (13) are more than 99% accurate for TC and more than 97% accurate"* *I honestly believe that this is where the author makes a big mistake because this approach has important consequences. The CT/AT ratio is key in his approximation. It is true that for CT/AT ratios<0.9 the approximation is quite correct and the van't Hoff model would work well, but for CT/AT values>0.9 equation 11 is going to present very low carbonate concentration (even lower than CO$_2$ concentrations and with high pCO$_2$ values generating very high biases when van't Hoff model is applied). The strange thing is that this is not indicated until very late in the article and is somewhat overlooked.*

> I realise this must arise from things not having been presented clearly enough in the previous manuscript, but the reviewer's assertion that the approximation will not work does not stand up to the evidence presented in the manuscript that the approximation does in fact work rather well, such as being able to very closely reproduce the t-$f\mathrm{CO}_2$ relationship calculated with CO2SYS, with less than 1 µatm RMSD in $f\mathrm{CO}_2$ for over 97% of the global surface ocean. There is no evidence for the 'big mistake' suggested by the reviewer.

> As mentioned in the manuscript, the approximations have been previously used successfully for getting a first-order understanding of a closely related marine carbonate system problem (Humphreys et al., 2018), equivalently to how they are being used here.

> The corresponding section has been expanded for clarity and brought forward to the very start of the R&D (Sec. 3.1.3 and Fig. 3 have become Sect. 3.1.1 and Fig. 1).

*Line 300-303. "At each grid point, we computed the mean across all years separately for each month for temperature, salinity, AT and TC. We then used PyCO2SYS to calculate fCO2 from these variables at 50 evenly spaced temperatures from –1.8 to 35.83 °C (i.e., the range of the OceanSODA-ETZH data product) at each month and grid point. Next, we fit Eq. (19) to the generated t and fCO2 data to find the best fitting bh value at each point." To be the key part of the way to obtain the equation 35 I think it is poorly explained.*

> I have expanded the explanation in the text to make this clearer.

*Especially because the equation first obtains 50 $f\mathrm{CO}_2$ data by applying the $\mathrm{CO}_2$SYS equations, and then adjusting it to the van`t Hoff equation which would only be applicable with high precision with low CT/AT values, i.e. with low $f\mathrm{CO}_2$ values which implies a low error in its adjustment. On the contrary, for high CT/AT values (high $f\mathrm{CO}_2$ values) the application of equation 19 already deviates from equation 11, and the biases generated amplify the errors in the $f\mathrm{CO}_2$ estimates using bh. (Fig 3b).*

> I am not certain what the reviewer is trying to say here. Yes, the approximation is worse at the higher CT/AT values that occur in less than 3% of the global surface ocean, and this is discussed in Sect. 3.1.1 and illustrated in Fig. 1 (which were Sect. 3.1.3 and Fig. 3 of the original manuscript).

*Line 305-66 "The coefficients and their variance-covariance matrix are provided in the Supplementary Information (Supp. Tables 1-2)." It would be more informative for the reader to include the uncertainties of each of the coefficients and their level of significance.*

> The uncertainties in the coefficients are already provided in Supp. Table 2. For the level of significance, I added an extra column to Supp. Table 1, which contains the coefficient values for normalised predictors, and added a note to the caption explaining that the relative magnitudes of these normalised coefficients show the relative importance of each coefficient to the fit.

> During this process I noticed that values for coefficients $u_2$-$u_5$ were not entered into Supp. Tables 1 and 2 in the correct order, so this has also been corrected.

*Line 352. Legend Figure 1 "a) Variation of fCO2 with temperature according to the measurements of Takahashi et al. (1993)". The legend is misleading. The variations of*

*fCO₂ with temperature are not described, but the anomaly of $fCO_2$ with respect to that estimated using various parameterizations, including the two proposed by Takahashi et al. 1993.*

➢ The rest of the partially quoted sentence resolves the misunderstanding here, as it states, "… all normalised to the linear fit". I have moved this phrase to earlier in the sentence to make it more obvious.

*Line 413. Figure 2. How can it be explained that if the coefficient obtained from van't Hoff called 'bh fitted' shows a behavior so different, and worse, from the bh parameterized van't Hoff coefficient considering that it is derived from that one.*

➢ The '$b_h$ fitted' curve uses a single $b_h$ value as fitted to the Takahashi et al. (1993) dataset globally. This does not account for variability of $b_h$ with the hydrographic conditions. The '$b_h$ parameterised' curve uses a variable $b_h$ value based on the parameterisation in Eq. (35).

*And on the other hand, the proposed parameterized bh enhancement is no better than the application of uLu00 as proposed by Wanninkof et al 2022.*

➢ The parameterised $b_h$ is indeed equal to or slightly better (closer to zero) than Lu00 throughout the $\Delta t$ range. But the method of Wanninkhof et al. requires a second marine carbonate system variable.

*Line 433 Figure 3. Figure 3a clearly shows that the proposed estimate (bh van`t Hoff) of fCO2 changes due to temperature changes does not improve on the more accurate alternative of direct application of CO₂SYS when the CT/AT ratio>0.95.*

➢ Correct, and this was stated in lines 429-430 of the originally submitted manuscript and is discussed in Sect. 3.1.1 of the revised manuscript. Again, the aim is not to improve on CO2SYS, but to find something that approaches the same accuracy without needing a second carbonate system variable.

*The legend to Figure 3b shows the RMSE of bh, but the units is K-1 and not the erroneously written µatm-1.*

➢ The unit of the RMSD is indeed µatm as written (not µatm$^{-1}$ and not K$^{-1}$). This is the same as for the following reviewer comment.

*Line 435 "the bh fit is less than 1 µatm". The bh unit is J mol -1 such as is indicated in Line 202.*

➢ As for the previous reviewer comment, the unit is correct, the RMSD is for $fCO_2$ calculated with the $b_h$ fit – sentence updated to clarify.

*Line 449 "parameterisations. However, while these issues might contribute a component of the discrepancy – i.e., the main pattern with RMSD ~1 µatm seen for TC/AT less than ~0.95 – there is no reason to expect their influence to be correlated with TC/AT, so they cannot be the entire explanation." How is it not possible that the author himself seems unaware of the limitations of equation 19 which proceeds from a strong simplification of equation 11? Just at CT/AT values below ~0.95, the term removed from equation 1 becomes determinant because the denominator tends to zero. We are in the environment*

*of the first equivalence point where carbonate concentrations are equal to $CO_2$ concentrations.*

➢ The author is well aware of these limitations, as they were discussed in the very next sentence in the manuscript (which is quoted by the reviewer directly below: lines 452-457). The corresponding section has been revised for improved clarity.

*Line 452-457 "Inaccuracies in the approximations Ax and/or Tx, used in generating Eq. (19), likely also play a role at higher TC/AT. The fraction of TC comprised of $[CO_2(aq)]$, which is ignored in Tx, increases with TC/AT, while the fraction of non-carbonate alkalinity, ignored in Ax, decreases with increasing TC/AT. Consequently, the approximation that $[HCO3–]2/[CO32-]$ is constant across different temperatures (Eq. 16), which emerges from the definitions of Ax and Tx (Eqs. 12-15), becomes less accurate with increasing TC/AT. The Tx approximation may be the problem here rather than Ax, because the RMSD of the bh fit is positively correlated with the error in Tx but negatively correlated with the error in Ax (Supp. Fig. 3)." Clearly the problem is the denominator of equation 11, and certainly it is the 'Inaccuracies in the approximations Ax and/or Tx, used in generating Eq. (19),' There is not the slightest doubt...*

➢ Up to here, the reviewer is correct (and is agreeing with the manuscript)...

*... and this calls into question the usefulness of bh and in the background of this whole article.*

➢ ... but this is not a valid conclusion. This is essentially the same point as the reviewer made earlier on line 177, so please refer to that response.

*Why use an alternative parameterization to avoid using CO2SYS which is always going to be less accurate even if we have indeterminacies in the equilibrium constants?*

➢ Alternatives to solving the complete marine carbonate system are useful because $fCO_2$ measurements are often not accompanied by a second carbonate system parameter.

➢ The reviewer's statement that the approach presented here is 'always going to be less accurate' is unfounded and currently false; uncertainties in the equilibrium constants significantly increase the uncertainty in carbonate system calculations; see my reply to the reviewer's comment on lines 98-99.

*Line 520 "But we now know that u should follow a particular curvature that can be represented with only one adjustable parameter (bh)" Sure? The equation 19 is a simplification of equation 11.*
*On the one hand, the equation 11 contains more factors than the apparent equilibrium constants of the marine carbonate equilibrium and therefore does not have to follow exactly the van't Hoff equation. On the other hand, the apparent (or empirical) carbonic acid constants (K1 and K2) as well as the CO2 saturation (Ko) contain more summands than the one given by the van't Hoff equation which are polynomial functions of the kelvin temperature.*

➢ Yes, here the phrasing was not clear that I am referring to the first-order form of u, which can be modelled well with equation 19. Added "to first order" to the sentence to clarify this.

*Line 575 (Figure 5) and594-597. "The SB21 parameterisation (Schockman and Byrne, 2021) consists of new, spectrophotometric measurements of the product K1\*K2\* which …k2\*, which resulted in overall virtually zero variability in total υ. This low variability in total υ is echoed by the Su20 parameterisation (Sulpis et al., 2020), which is based on field observations where AT, TC, pH and fCO2 were measured simultaneously, but the low variability is arrived at in a different way, with rather different distributions for the individual K1\* and K2\* effects". Interesting figure.*

*Obviously, the probability curves represent the spatial distribution of the ocean surface and this is strongly dependent on latitude. But looking at a relative perspective when comparing one set of constants with others, it is interesting to note that both the Schockman &Byrne 'Mehrbach' option (SB21) and Sulpis 2020 (Su20) show virtually no spatial variability with values very close to that estimated by Takahashi et al. (1993). This is an interesting aspect of this study.*

> ➤ This is interesting (and indeed already mentioned in the manuscript, lines 594-599 of the original), although I'm not sure that it delivers any new insight. Neither of these parameterisations fits the Takahashi et al. (1993) dataset very well (Supp. Fig. 2). I added a note to emphasise this point to the section.

*Line 664 Conclusions. This epigraph is too long, and in some parts, it is rather a new discussion.*

> ➤ It has been shortened with some parts relocated to other sections.

*References*

*Rodgers, K. B., Schwinger, J., Fassbender, A. J., Landschützer, P., Yamaguchi, R., Frenzel, H., et al. (2023). Seasonal variability of the surface ocean carbon cycle: a synthesis. Global Biogeochemical Cycles,37, e2023GB007798. https://doi.org/10.1029/2023GB007798*

Humphreys, M. P., Daniels, C. J., Wolf-Gladrow, D. A., Tyrrell, T., and Achterberg, E. P.: On the influence of marine biogeochemical processes over $CO_2$ exchange between the atmosphere and ocean, Mar. Chem., 199, 1–11, https://doi.org/10.1016/j.marchem.2017.12.006, 2018.

Orr, J. C., Epitalon, J.-M., Dickson, A. G., and Gattuso, J.-P.: Routine uncertainty propagation for the marine carbon dioxide system, Mar. Chem., 207, 84–107, https://doi.org/10.1016/j.marchem.2018.10.006, 2018.

---

## Author Comment (AC2)

*Matthew Humphreys provides a physical chemical basis to the frequently used empirical relationship of fCO2 with temperature along with an thorough uncertainty analysis. He has incorporated the results into the PyCO2SYS software. The temperature relationship is frequently used to correct surface water fCO2 measurements to in situ temperatures. Based on the uncertainty analysis presented, the uncertainty for this correction can be reduced from 0.24 % to 0.06 %.*

➢ I am grateful to the reviewer for taking the time to carefully read the manuscript and provide thoughtful and constructive feedback.

*The manuscript is exhaustive and laid out in a coherent fashion. It is well-referenced and thoroughly researched. Non-the-less it is a challenging read and final results and conclusions are not always clearly laid out.*

➢ Thank you for these comments. I have rewritten many of the trickier parts of the paper to try to make them easier to follow.

*For instance, the equations to determine the temperature dependence are not that clearly laid out. That is, it would be useful if an additional equation was added after Eqns. 19-21 where the numerical values constants for $b_h$ and R were included.*

➢ I am reluctant to add a series of equations with the numerical values for $b_h$ filled in because there would be potentially so many of them. There is the theoretical $b_h$ value, the $b_h$ value fitted to the Takahashi et al. (1993) dataset, now also the $b_h$ value fitted to the Lee and Millero (1995) dataset, and a continuous spectrum of possible $b_h$ values from the parameterisation. All the equations would be identical except for that value. The value for R is also prone to occasional revision. That said, I think that the underlying problem here may be that it's difficult for the reader to keep track of all the different $b_h$ values. As such, I've added a table (Table 1) to assist with this, which hopefully fixes the problem in a different way.

*A central premise of the work is that the proper functionality for the temperature dependence is 1/T, following the van t'Hoff relationship, rather than t which is well-founded but the fit of a 1/T relationship to the experimental data of Takahashi et al. 1993 is worse than a linear fit with t. At the end of the manuscript the author correctly states that over a narrow range the functionality of 1/T versus t is very similar.*

➢ The fit to 1/T indeed has a higher (i.e., worse) RMSD than the fit to T. But the 1/T fit is still not inconsistent with the uncertainty in the measurements. If multiple possible fits (linear, quadratic, 1/T) all fall within the uncertainty window, then especially when there are so few data points to compare against, I don't think it's really valid to simply say that the one with the lowest RMSD must be 'correct'. See also related comment below on lines 335-340.
➢ Nevertheless, the revised manuscript takes a more critical view of the agreement (or lack thereof) between the new form and the Takahashi et al. (1993) measurements, thanks in a large part to this reviewer's suggestion to also look at Lee and Millero (1995).

*As pointed out by others, the temperature dependence will depend on the bicarbonate and carbonate concentrations of the seawater. This is also described in this manuscript (eqn12-18)*

*but never emphasized. The important point is that no single simple equation can predict fCO2 solely based on temperature but that information on TA and DIC needs to be implicitly included. It might be worth emphasizing that knowledge or estimate of TA and DIC will decrease the uncertainty in the ln(fCO2-T) relationship.*

➢ Yes, I think this is an important point that was not made clearly enough in the preprint. This is emphasised more with a new paragraph towards the end of Section 3.1.3. Related to comment on line 455 below.

*One aspect that is not emphasized is why this empirical single fCO2-temperature dependence developed by Takahashi based on a single seawater composition "works" for the surface ocean as described in Wanninkhof et al. 2022:" The theoretical temperature dependence expressed as ∂ln(fCO2w)/∂T = B0 + B1 T shows a stronger dependency at lower temperatures, and weaker dependency at low TA/DIC values (Fig. 4). These conditions often go hand-in-hand in the surface ocean. That is, surface waters of the world's oceans have lower temperatures and lower TA/DIC at higher latitudes such that the two factors will oppose each other."*

➢ This is indeed an interesting point. A new figure (Fig. 7) and an additional paragraph of text at the end of Sect. 3.3.1 have been added to discuss it further.

*For experimental data and verification the paper relies heavily on the pCO2-temperature relationship of Takahashi that was derived from 1 seawater sample and 8 measurements at 7 temperatures from 2-25 C. The paper also assumes that deviations from a linear trend are likely at higher temperatures and suggests, correctly, that more systematic studies have to be undertaken. While several groups have undertaken such studies, they have failed to publish it in the open literature. A notable exception is that of Lee and Millero (1995) that provides measurements from 5-30 C and obtains a linear dependence very similar to the of Takahashi et (1993) (see attached figure) particularly if the sample at 5 °C is omitted that is questionable (Lee personal comm, 1996). Of note are that the 30 C value falls in line with other points, but that the temperature dependence using a polynomial fit is twice that of Takahashi.*

➢ Thank you for bringing this dataset to my attention. It has been incorporated in the manuscript and leads to my taking a more critical view of the agreement between the model and these measurements. See also related comment on line 331.
➢ However, it is worth mentioning that the Lee and Millero (1995) measurements don't seem to be consistent with anything other than Takahashi et al. (1993), and because of the different TC/AT they shouldn't be as consistent with Takahashi et al. (1993) as they are, so they have been rather tricky to integrate into the analysis.

*Comments by line number*

*Line 11: omit "purely"*

➢ Done.

*Line 25 and beyond : Besides the constants the concentrations of the species will impact the relationship as well*

➢ Updated text.

*Line 47: global mean ΔfCO2 ≈ -5 µatm (Fay et al., 2024)*

➢ Thank you! Reference added and value updated.

*Line 50 and beyond: "minimum accuracy of 0.5 %. " since ΔfCO2 is often the quantity of interest it is commonly expressed as < 2 µatm for "climate quality" rather than a %*

➢ Added equivalent µatm value here. I have retained the 0.5% too because this is how it is 'officially' expressed by Newton et al.

*Line 55: state that the surface seawater is equilibrated with an enclosed headspace and the headspace is measured. That is fCO2 is fundamentally a gas phase property*

➢ Text updated.

*Line 59: This criterium is for SOCAT dataset flags of A and B; SOCAT accepts all fCO2 measurements*

➢ Text updated.

*Line 65: Of note is that fCO2 is also measured on discrete samples at fixed temperature usually 20 ℃ by select groups. In these case conversion to fCO2 at in situ temperatures is much greater.*

➢ Added this example to the list.

*Line 73: replace: "measured" by "reported"*

➢ Done.

*Line 93: insert "υ, " after "calculating"*

➢ Done.

*Line 109: Note that Takahashi et al. 2009 provide the equation as well including the integrated form*

➢ I'm not sure I understand how this comment fits with the text here, so I haven't changed anything.

*Sections 2.1 and 2.3: Very nice description of physical basis and uncertainties*

➢ Thank you!

*Line 139: note that 8 measurements were taken at 7 temperatures.*

➢ Done.

*Line 331: Note Lee and Millero (1996) have a measurement at 30 C that could be used to spot-check the difference between υl and υq above about 25 C*

➢ Thanks. See response to comment above "For experimental data and verification…"

*Line 335-340: the issue that the proposed fit does not do as well as the original is a significant point, even if both fall within the calculated uncertainty. Again the Lee and Millero (1996) measurements could shed some light on the fundamental issue if the proposed equations have shortcomings.*

➢ Agreed. See response to comment above "A central premise of the work…"

*Line 427: Section 3.1.3 I found this section confusing in part because the previous sections discuss fitting with experimental data while this section discusses using the Lueker constants. I had expected that the author would use the experimental data presented in Table 3 of the paper of Lueker et al. which would be a good test of their parameterizations as the tests were done at several temperatures and varying fCO2, TA and DIC.*

➢ It's not really possible to perform this analysis using the suggested experimental data directly, because in order to test the parameterisation we need $f\mathrm{CO_2}$ values for multiple temperatures where salinity, DIC and alkalinity are all held constant. In the Lueker et al. dataset, DIC is rather different for every measurement, and there is some variability in salinity and alkalinity too. I have updated the explanation of this section and moved it to an earlier place in the R&D to try to make it flow better and less confusing.

*Line 455: This point could be empathized "Consequently, the approximation that [HCO3–]2/[CO32–] is constant across different temperatures (Eq. 16), which emerges from the definitions of Ax and Tx (Eqs. 12-15), becomes less accurate with increasing TC/AT.*

➢ I have reworked this text and emphasised this point more. See also response to comment above "As pointed out by others…"

*Line 675 and before:" but this does not account for variability in υ through space and time" [while this is explained in the introduction], strictly speaking υ varies with temperature and chemical composition, not space and time.*

➢ Agreed, text updated.

*Lee, K., and F. J. Millero, 1995: Thermodynamic studies of the carbonate system in seawater. Deep-Sea Res., **42,** 2035-2061.*

*Takahashi, T., and Coauthors, 2009: Climatological mean and decadal change in surface ocean pCO2, and net sea-air CO2 flux over the global oceans. Deep -Sea Res II, **2009,** 554-577.*

---

## Author Response (AR2)

*Dear Matthew,*

*Thanks for the revised manuscript, which is much improved, also due to the help of the very useful comments by both referees. I am satisfied with your responses to the reviewers and the new version. There are only some final technical issues that I have listed below.*

➢ Thank you for your support for this manuscript and the comments which have also helped tidy up a few loose ends.

*L137 Earlier the unit of v was given as % °C-1. Please be consistent. And please write: in units of per temperature (note: x-1 only for symbols)*

➢ Changed to % °C$^{-1}$

*L286 It would be helpful to spent one sentence on what PyCO2SYS actually is and stands for. Not every reader may be familiar with this software.*

➢ Added a short paragraph to the start of section 2.3:
"PyCO2SYS is a free and open source Python package which can be used to solve the marine carbonate system, that is, to calculate the equilibrium balance of the main acid-base systems in seawater and related properties (Humphreys et al., 2022). PyCO2SYS was originally based on the MATLAB/GNU Octave program CO2SYS.m (v2.0.5; Orr et al., 2018), itself part of a family of similar software tools beginning with the original CO2SYS for MS-DOS (Lewis and Wallace, 1998)."

*L328 It would be good to give some info on the data set. There are several other data sets available as well, and therefore the choice for this one should be justified.*

➢ Added a couple of sentences to the start of section 2.4:
"OceanSODA-ETZH provides fields of several marine carbonate system and other hydrographic parameters gridded across the surface ocean (1° × 1°) and through time (monthly from 1985 to 2018). The parameter fields were generated using an ensemble cluster-regression approach based on observations in SOCAT (Bakker et al., 2016) and GLODAP (Lauvset et al., 2016). Other similar data products exist, but the main patterns and variability in surface ocean carbonate chemistry are well enough constrained that the choice of a particular data product will not significantly affect the global-scale and time-averaged analyses conducted here."

*L328 Please use date format 4 December 2023 throughout the paper*

➢ Done

*L364 It is somewhat confusing that eq. 19 is mentioned for determining vh , as vh does not occur in eq. 19. Please change wording to make this clear. It also occurs at other places following.*

➢ Some (but not all) of these should probably be referring to Eq. (20). Have updated, and added a note in Sect. 3.1.1 that (19) is the integrated form of (20):

"Equation (20) cannot be fitted directly to $fCO_2$ data because $u$ represents the derivative with respect to temperature; instead, we need to fit Eq. (19) to obtain the unknown $b_h$ which can then be used in Eq. (20) for $u_h$."

*L390 delete one of the two "that"*
*L450 delete slightly, as this is a difference of 25*
> ➤ Both done

*I have one suggestion which you may consider applying. The theory and reasoning in this manuscript is not easy stuff (it is a challenging read, as one of the referees mentioned it) and the paper is quite long. Therefore, you may consider to write in the conclusions section, an instruction of what to do if one wants to adjust the temperature of pCO2 measurements to the in situ temperature. In the Takahashi paper everyone can simply see how to do this. It would be good when such a clear instruction would also appear here.*
> ➤ I added a paragraph to the start of the Conclusions with an instruction of specifically how to do this, which equations need to be used:
> "Seawater $fCO_2$ data can be adjusted to different temperatures using either Eq. (2) or Eq. (4) (which are mathematically identical) with an appropriate expression for $Y$, which is $u$ integrated over the temperature range of interest. The new approach proposed here uses $Y_h$ as defined in Eq. (21), which has one unknown coefficient ($b_h$). The value of $b_h$ can be found by fitting to experimental data where available (e.g., Table 1), but spatiotemporal variability in $b_h$ should be accounted for, for example with the parameterisation in Eq. (35). This approach, as well as the earlier linear and quadratic approaches of Takahashi et al. (1993), have been built into the PyCO2SYS software as of v1.8.3."

*Thanks and with best wishes*
*Mario*

*Additional private note (visible to authors and reviewers only):*
*This is a relatively long paper. To increase the readability, I would encourage to shorten where possible. I realize that this may be difficult, but maybe you see some possibilities. If you think this is not possible, I am fine with it as well. I just think that the paper will be read better when it is not too long.*
> ➤ I agree in principle and did read through the manuscript with this in mind, but to reduce the length by any significant amount now would require a substantial amount extra work to ensure that everything still flows and is complete, which is going to be very difficult to fit in with other commitments at the present time, so I have not made any cuts here.

Additional changes:
> ➤ Updated GLODAP citation to the most recent version (Lauvset et al., 2022 => Lauvset et al., 2024).